# DPOK: Reinforcement Learning for Fine-tuning Text-to-Image Diffusion Models

**Ying Fan**[*,1,2], **Olivia Watkins**[3], **Yuqing Du**[3], **Hao Liu**[3], **Moonkyung Ryu**[1], **Craig Boutilier**[1], **Pieter Abbeel**[3], **Mohammad Ghavamzadeh**[†,4], **Kangwook Lee**[2], **Kimin Lee**[*,†,5]

[*]Equal technical contribution     [†]Work was done at Google Research

[1]Google Research    [2]University of Wisconsin-Madison    [3]UC Berkeley    [4]Amazon    [5]KAIST

## Abstract

Learning from human feedback has been shown to improve text-to-image models. These techniques first learn a reward function that captures what humans care about in the task and then improve the models based on the learned reward function. Even though relatively simple approaches (e.g., rejection sampling based on reward scores) have been investigated, fine-tuning text-to-image models with the reward function remains challenging. In this work, we propose using online reinforcement learning (RL) to fine-tune text-to-image models. We focus on *diffusion models*, defining the fine-tuning task as an RL problem, and updating the pre-trained text-to-image diffusion models using policy gradient to maximize the feedback-trained reward. Our approach, coined DPOK, integrates policy optimization with KL regularization. We conduct an analysis of KL regularization for both RL fine-tuning and supervised fine-tuning. In our experiments, we show that DPOK is generally superior to supervised fine-tuning with respect to both image-text alignment and image quality. Our code is available at https://github.com/google-research/google-research/tree/master/dpok.

## 1 Introduction

Recent advances in *diffusion models* [10, 37, 38], together with pre-trained text encoders (e.g., CLIP [27], T5 [28]) have led to impressive results in text-to-image generation. Large-scale text-to-image models, such as Imagen [32], Dalle-2 [29], and Stable Diffusion [30], generate high-quality, creative images given novel text prompts. However, despite these advances, current models have systematic weaknesses. For example, current models have a limited ability to compose multiple objects [6, 7, 25]. They also frequently encounter difficulties when generating objects with specified colors and counts [12, 17].

*Learning from human feedback (LHF)* has proven to be an effective means to overcome these limitations [13, 17, 42, 43]. Lee et al. [17] demonstrate that certain properties, such as generating objects with specific colors, counts, and backgrounds, can be improved by learning a *reward function* from human feedback, followed by fine-tuning the text-to-image model using supervised learning. They show that simple supervised fine-tuning based on reward-weighted loss can improve the reward scores, leading to better image-text alignment. However, supervised fine-tuning often induces a deterioration in image quality (e.g., over-saturated or non-photorealistic images). This is likely due to the model being fine-tuned on a fixed dataset that is generated by a pre-trained model (Figure 1(a)).

In this work, we explore using *online reinforcement learning (RL)* for fine-tuning text-to-image diffusion models (Figure 1(b)). We show that optimizing the expected reward of a diffusion model's image output is equivalent to performing policy gradient on a multi-step diffusion model under certain

37th Conference on Neural Information Processing Systems (NeurIPS 2023).

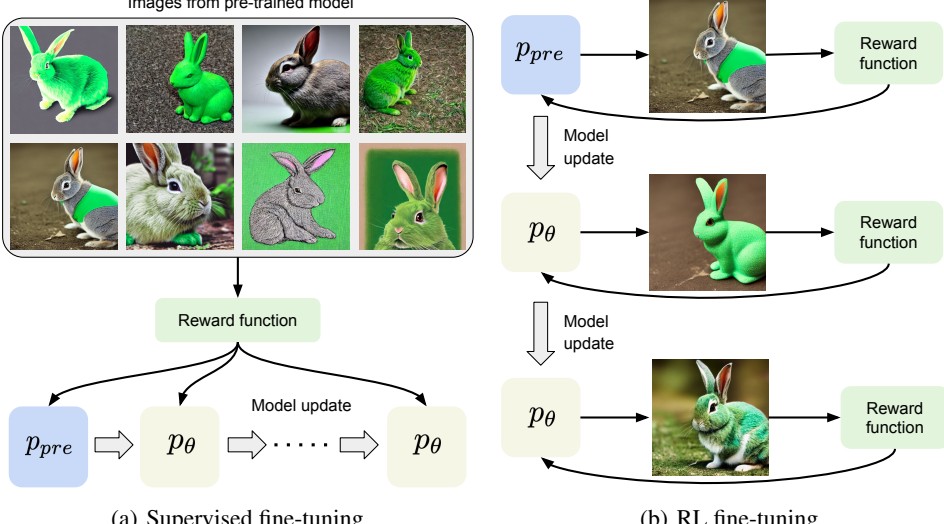

Figure 1: Illustration of (a) reward-weighted supervised fine-tuning and (b) RL fine-tuning. Both start with the same pre-trained model (the blue rectangle). In supervised fine-tuning, the model is updated on a fixed dataset generated by the pre-trained model. In contrast, the model is updated using new samples from the previously trained model during online RL fine-tuning.

regularity assumptions. We also incorporate Kullback–Leibler (KL) divergence with respect to the pre-trained model as regularization in an online manner, treating this as an implicit reward.

In our experiments, we fine-tune the Stable Diffusion model [30] using ImageReward [43], an open-source reward model trained on a large dataset comprised of human assessments of (text, image) pairs. We show that online RL fine-tuning achieves strong text-image alignment while maintaining high image fidelity by optimizing its objective in an online manner. *Crucially, online training allows evaluation of the reward model and conditional KL divergence beyond the (supervised) training dataset.* This offers distinct advantages over supervised fine-tuning, a point we demonstrate empirically. In our empirical comparisons, we also incorporate the KL regularizer in a supervised fine-tuning method for a fair comparison.

Our contributions are as follows:

- We frame the optimization of the expected reward (w.r.t. an LHF-reward) of the images generated by a diffusion model given text prompts as an online RL problem. Moreover, we present DPOK: **D**iffusion **P**olicy **O**ptimization with **K**L regularization, which utilizes KL regularization w.r.t. the pre-trained text-to-image model as an implicit reward to stabilize RL fine-tuning.
- We study incorporating KL regularization into supervised fine-tuning of diffusion models, which can mitigate some failure modes (e.g., generating over-saturated images) in [17]. This also allows a fairer comparison with our RL technique.
- We discuss the key differences between supervised fine-tuning and online fine-tuning of text-to-image models (Section 4.3).
- Empirically, we show that online fine-tuning is effective in optimizing rewards, which improves text-to-image alignment while maintaining high image fidelity.

## 2    Related Work

**Text-to-image diffusion models.**    Diffusion models [10, 37, 39] are a class of generative models that use an iterative denoising process to transform Gaussian noise into samples that follow a learned data distribution. These models have proven to be highly effective in a range of domains, including image generation [4], audio generation [14], 3D synthesis [26], and robotics [3]. When combined with large-scale language encoders [27, 28], diffusion models have demonstrated impressive performance in text-to-image generation [29, 30, 32]. However, there are still many known weaknesses of existing text-to-image models, such as compositionality and attribute binding [6, 20] or text rendering [21].

**Learning from human feedback.** Human assessments of (or preferences over) learned model outcomes have been used to guide learning on a variety of tasks, ranging from learning behaviors [16] to language modeling [1, 23, 19, 41]. Recent work has also applied such methods to improve the alignment of text-to-image models. Human preferences are typically gathered at scale by asking annotators to compare generations, and a *reward model* is trained (e.g., by fine-tuning a vision-language model such as CLIP [27] or BLIP [18]) to produce scalar rewards well-aligned with the human feedback [43, 13]. The reward model is used to improve text-to-image model quality by fine-tuning a pre-trained generative model [17, 42]. Unlike prior approaches, which typically focus on reward-filtered or reward-weighted supervised learning, we develop an online fine-tuning framework with an RL-based objective.

**RL fine-tuning of diffusion models.** Fan & Lee [5] first introduced a method to improve pre-trained diffusion models by integrating policy gradient and GAN training [8]. They used policy gradient with reward signals from the discriminator to update the diffusion model and demonstrated that the fine-tuned model can generate realistic samples with few diffusion steps with DDPM sampling [10] on relatively simple domains (e.g., CIFAR [15] and CelebA [22]). In this work, we explore RL fine-tuning especially for large-scale text-to-image models using human rewards. We also consider several design choices like adding KL regularization as an implicit reward, and compare RL fine-tuning to supervised fine-tuning.

Concurrent and independent from our work, Black et al. [2] have also investigated RL fine-tuning to fine-tune text-to-image diffusion models. They similarly frame the fine-tuning problem as a multi-step decision-making problem, and demonstrate that RL fine-tuning can outperform supervised fine-tuning with reward-weighted loss [17] in optimizing the reward, which aligns with our own observations. Furthermore, our work analyzes KL regularization for both supervised fine-tuning and RL fine-tuning with theoretical justifications, and shows that adopting KL regularization is useful in addressing some failure modes (e.g., deterioration in image quality) of fine-tuned models. For a comprehensive discussion of prior work, we refer readers to Appendix D.

## 3 Problem Setting

In this section, we describe our basic problem setting for text-to-image generation of diffusion models.

**Diffusion models.** We consider the use of *denoising diffusion probabilistic models (DDPMs)* [10] for image generation and draw our notation and problem formulation from [10]. Let $q_0$ be the data distribution, i.e., $x_0 \sim q_0(x_0)$, $x_0 \in \mathbb{R}^n$. A DDPM approximates $q_0$ with a parameterized model of the form $p_\theta(x_0) = \int p_\theta(x_{0:T}) dx_{1:T}$, where $p_\theta(x_{0:T}) = p_T(x_T) \prod_{t=1}^T p_\theta(x_{t-1}|x_t)$ and the *reverse process* is a Markov chain with the following dynamics:

$$p(x_T) = \mathcal{N}(0, I), \qquad p_\theta(x_{t-1}|x_t) = \mathcal{N}\big(\mu_\theta(x_t, t), \Sigma_t\big). \tag{1}$$

A unique characteristic of DDPMs is the exploitation of an approximate posterior $q(x_{1:T}|x_0)$, known as the *forward* or *diffusion process*, which itself is a Markov chain that adds Gaussian noise to the data according to a variance schedule $\beta_1, \ldots, \beta_T$:

$$q(x_{1:T}|x_0) = \prod_{t=1}^T q(x_t|x_{t-1}), \qquad q(x_t|x_{t-1}) = \mathcal{N}(\sqrt{1-\beta_t}\, x_{t-1}, \beta_t I). \tag{2}$$

Let $\alpha_t = 1 - \beta_t$, $\bar{\alpha}_t = \prod_{s=1}^t \alpha_s$, and $\tilde{\beta}_t = \frac{1-\bar{\alpha}_{t-1}}{1-\bar{\alpha}_t}\beta_t$. Ho et al. [10] adopt the parameterization $\mu_\theta(x_t, t) = \frac{1}{\sqrt{\alpha_t}}\left(x_t - \frac{\beta_t}{\sqrt{1-\bar{\alpha}_t}}\epsilon_\theta(x_t, t)\right)$.

Training a DDPM is performed by optimizing a variational bound on the negative log-likelihood $\mathbb{E}_q[-\log p_\theta(x_0)]$, which is equivalent to optimizing:

$$\mathbb{E}_q\left[\sum_{t=1}^T \text{KL}\big(q(x_{t-1}|x_t, x_0)\|p_\theta(x_{t-1}|x_t)\big)\right]. \tag{3}$$

Note that the variance sequence $(\beta_t)_{t=1}^T \in (0, 1)^T$ is chosen such that $\bar{\alpha}_T \approx 0$, and thus, $q(x_T|x_0) \approx \mathcal{N}(0, I)$. The covariance matrix $\Sigma_t$ in (1) is often set to $\sigma_t^2 I$, where $\sigma_t^2$ is either $\beta_t$ or $\tilde{\beta}_t$, which is not trainable. Unlike the original DDPM, we use a latent diffusion model [30], so $x_t$'s are latent.

**Text-to-image diffusion models.** Diffusion models are especially well-suited to *conditional* data generation, as required by text-to-image models: one can plug in a classifier as guidance function [4], or can directly train the diffusion model's conditional distribution with classifier-free guidance [9].

Given text prompt $z \sim p(z)$, let $q(x_0|z)$ be the data distribution conditioned on $z$. This induces a joint distribution $p(x_0, z)$. During training, the same noising process $q$ is used regardless of input $z$, and both the unconditional $\epsilon_\theta(x_t, t)$ and conditional $\epsilon_\theta(x_t, t, z)$ denoising models are learned. For data sampling, let $\bar{\epsilon}_\theta = w\epsilon_\theta(x_t, t, z) + (1 - w)\epsilon_\theta(x_t, t)$, where $w \geqslant 1$ is the guidance scale. At test time, given a text prompt $z$, the model generates conditional data according to $p_\theta(x_0|z)$.

## 4 Fine-tuning of Diffusion Models

In this section, we describe our approach for online RL fine-tuning of diffusion models. We first propose a Markov decision process (MDP) formulation for the denoising phase. We then use this MDP and present a policy gradient RL algorithm to update the original diffusion model. The RL algorithm optimizes an objective consisting of the reward and a KL term that ensures the updated model is not too far from the original one. We also present a modified supervised fine-tuning method with KL regularization and compares it with the RL approach.

### 4.1 RL Fine-tuning with KL Regularization

Let $p_\theta(x_{0:T}|z)$ be a text-to-image diffusion model where $z$ is some text prompt distributed according to $p(z)$, and $r(x_0, z)$ be a reward model (typically trained using human assessment of images).

**MDP formulation:** The denoising process of DDPMs can be modeled as a $T$-horizon MDP:

$$s_t = (z, x_{T-t}), \quad a_t = x_{T-t-1}, \quad P_0(s_0) = \big(p(z), \mathcal{N}(0, I)\big), \quad P(s_{t+1} \mid s_t, a_t) = (\delta_z, \delta_{a_t}),$$

$$R(s_t, a_t) = \begin{cases} r(s_{t+1}) = r(x_0, z) & \text{if } t = T - 1, \\ 0 & \text{otherwise.} \end{cases}, \quad \pi_\theta(a_t \mid s_t) = p_\theta(x_{T-t-1} \mid x_{T-t}, z), \quad (4)$$

in which $s_t$ and $a_t$ are the state and action at time-step $t$, $P_0$ and $P$ are the initial state distribution and the dynamics, $R$ is the reward function, and $\pi_\theta$ is the parameterized policy. As a result, optimizing policy $\pi_\theta$ in (4) is equivalent to fine-tuning the underlying DDPM.[1] Finally, we denote by $\delta_z$ the Dirac distribution at $z$.

It can be seen from the MDP formulation in (4) that the system starts by sampling its initial state $s_0$ from the Gaussian distribution $\mathcal{N}(0, I)$, similar to the first state of the dinoising process $x_T$. Given the MDP state $s_t$, which corresponds to state $x_{T-t}$ of the denoising process, the policy takes the action at time-step $t$ as the next denoising state, i.e., $a_t = x_{T-t-1}$. As a result of this action, the system transitions deterministically to a state identified by the action (i.e., the next state of the denoising process). The reward is zero, except at the final step in which the quality of the image at the end of the denoising process is evaluated w.r.t. the prompt, i.e., $r(x_0, z)$.

A common goal in re-training/fine-tuning the diffusion models is to maximize the expected reward of the generated images given the prompt distribution, *i.e.*,

$$\min_\theta \ \mathbb{E}_{p(z)}\mathbb{E}_{p_\theta(x_0|z)}[-r(x_0, z)]. \tag{5}$$

The gradient of this objective function can be obtained as follows:

**Lemma 4.1** (A modification of Theorem 4.1 in [5])**.** *If $p_\theta(x_{0:T}|z)r(x_0, z)$ and $\nabla_\theta p_\theta(x_{0:T}|z)r(x_0, z)$ are continuous functions of $\theta$, then we can write the gradient of the objective in (5) as*

$$\nabla_\theta \mathbb{E}_{p(z)}\mathbb{E}_{p_\theta(x_0|z)}[-r(x_0, z)] = \mathbb{E}_{p(z)}\mathbb{E}_{p_\theta(x_{0:T}|z)}\left[-r(x_0, z)\sum_{t=1}^T \nabla_\theta \log p_\theta(x_{t-1}|x_t, z)\right]. \tag{6}$$

*Proof.* We present the proof in Appendix A.1. □

---

[1]We keep the covariance in DDPM as constant, $\Sigma_t = \Sigma$, and only train $\mu_\theta$ (see Eq. 1). This would naturally provide a stochastic policy for online exploration.

---

**Algorithm 1** DPOK: Diffusion policy optimization with KL regularization

---

**Input**: reward model $r$, pre-trained model $p_{\text{pre}}$, current model $p_\theta$, batch size $m$, text distribution $p(z)$

    Initialize $p_\theta = p_{\text{pre}}$
    **while** $\theta$ not converged **do**
        Obtain $m$ i.i.d. samples by first sampling $z \sim p(z)$ and then $x_{0:T} \sim p_\theta(x_{0:T}|z)$
        Compute the gradient using Eq. (9) and update $\theta$
    **end while**
**Output**: Fine-tuned diffusion model $p_\theta$

---

Equation (6) is equivalent to the gradient used by the popular policy gradient algorithm, REINFORCE, to update a policy in the MDP (4). The gradient in (6) is estimated from trajectories $p_\theta(x_{0:T}|z)$ generated by the current policy, and then used to update the policy $p_\theta(x_{t-1}|x_t, z)$ in an online fashion.

Note that REINFORCE is not the only way to solve (5). Alternatively, one could compute the gradient through the trajectories to update the model; but the multi-step nature of diffusion models makes this approach memory inefficient and potentially prone to numerical instability. Consequently, scaling it to high-resolution images becomes challenging. For this reason, we adopt policy gradient to train large-scale diffusion models like Stable Diffusion [30].

**Adding KL regularization.**  The risk of fine-tuning purely based on the reward model learned from human or AI feedback is that the model may overfit to the reward and discount the "skill" of the initial diffusion model to a greater degree than warranted. To avoid this phenomenon, similar to [23, 41], we add the KL between the fine-tuned and pre-trained models as a regularizer to the objective function. Unlike the language models in which the KL regularizer is computed over the entire sequence/trajectory (of tokens), in text-to-image models, it makes sense to compute it only for the final image, i.e., $\text{KL}\big(p_\theta(x_0|z)\|p_{\text{pre}}(x_0|z)\big)$. Unfortunately, $p_\theta(x_0|z)$ is a marginal (see the integral in Section 3) and its closed-form is unknown. As a result, we propose to add an upper-bound of this KL-term to the objective function.

**Lemma 4.2.** *Suppose $p_{pre}(x_{0:T}|z)$ and $p_\theta(x_{0:T}|z)$ are Markov chains conditioned on the text prompt $z$ that both start at $x_T \sim \mathcal{N}(0, I)$. Then, we have*

$$\mathbb{E}_{p(z)}[\text{KL}(p_\theta(x_0|z))\|p_{pre}(x_0|z))] \leqslant \mathbb{E}_{p(z)}\left[\sum_{t=1}^{T}\mathbb{E}_{p_\theta(x_t|z)}\big[\text{KL}\big(p_\theta(x_{t-1}|x_t,z)\|p_{pre}(x_{t-1}|x_t,z)\big)\big]\right]. \quad (7)$$

We report the proof of Lemma 4.2 in Appendix A.2. Intuitively, this lemma tells us that the divergence between the two distributions over the output image $x_0$ is upper-bounded by the sum of the divergences between the distributions over latent $x_t$ at each diffusion step.

Using the KL upper-bound in (7), we propose the following objective for regularized training:

$$\mathbb{E}_{p(z)}\left[\alpha\mathbb{E}_{p_\theta(x_{0:T}|z)}[-r(x_0,z)] + \beta\sum_{t=1}^{T}\mathbb{E}_{p_\theta(x_t|z)}\big[\text{KL}\big(p_\theta(x_{t-1}|x_t,z)\|p_{\text{pre}}(x_{t-1}|x_t,z)\big)\big]\right], \quad (8)$$

where $\alpha, \beta$ are the reward and KL weights, respectively. We use the following gradient to optimize the objective (8):

$$\mathbb{E}_{p(z)}\mathbb{E}_{p_\theta(x_{0:T}|z)}\left[-\alpha r(x_0,z)\sum_{t=1}^{T}\nabla_\theta \log p_\theta(x_{t-1}|x_t,z) + \beta\sum_{t=1}^{T}\nabla_\theta\text{KL}\big(p_\theta(x_{t-1}|x_t,z)\|p_{\text{pre}}(x_{t-1}|x_t,z)\big)\right]. \quad (9)$$

Note that (9) has one term missing from the exact gradient of (8) (see Appendix A.3). Removing this term is for efficient training. The pseudo-code of our algorithm, which we refer to as DPOK, is summarized in Algorithm 1. To reuse historical trajectories and be more sample efficient, we can also use importance sampling and clipped gradient, similar to [36]. We refer readers to Appendix A.6 for these details.

## 4.2  Supervised Learning with KL Regularization

We now introduce KL regularization into supervised fine-tuning (SFT), which allows for a more meaningful comparison with our KL-regularized RL algorithm (DPOK). We begin with a supervised

fine-tuning objective similar to that used in [17], i.e.,

$$\mathbb{E}_{p(z)}\mathbb{E}_{p_{\text{pre}}(x_0|z)}[-r(x_0,z)\log p_\theta(x_0|z)]. \tag{10}$$

To compare with RL fine-tuning, we augment the supervised objective with a similar KL regularization term. Under the supervised learning setting, we consider $\text{KL}(p_{\text{pre}}(x_0|z)\|p_\theta(x_0|z))$, which is equivalent to minimizing $\mathbb{E}_{p_{\text{pre}}(x_0|z)}[-\log p_\theta(x_0|z)]$, given any prompt $z$. Let $q(x_t)$ be the forward process used for training. Since the distribution of $x_0$ is generally not tractable, we use approximate upper-bounds in the lemma below (see derivation in Appendix A.4).

**Lemma 4.3.** *Let $\gamma$ be the regularization weight and $\tilde{\mu}_t(x_t,x_0) := \frac{\sqrt{\bar{\alpha}_{t-1}}\beta_t}{1-\bar{\alpha}_t}x_0 + \frac{\sqrt{\alpha_t}(1-\bar{\alpha}_{t-1})}{1-\bar{\alpha}_t}x_t$. Assume $r(x_0,z) + \gamma > 0$, for all $x_0$ and $z$. Then, we have*

$$\mathbb{E}_{p(z)}\mathbb{E}_{p_{pre}(x_0|z)}[-(r(x_0,z)+\gamma)\log p_\theta(x_0|z)]$$
$$\leqslant \mathbb{E}_{p(z)}\mathbb{E}_{p_{pre}(x_0|z)}[(r(x_0,z)+\gamma)\sum_{t>1}\mathbb{E}_{q(x_t|x_0,z)}[\frac{1}{2\sigma_t^2}||\tilde{\mu}_t(x_t,x_0)-\mu_\theta(x_t,t,z)||^2]] + C_1. \tag{11}$$

*Moreover, we also have another weaker upper-bound in which $C_1$ and $C_2$ are two constants:*

$$\mathbb{E}_{p(z)}\mathbb{E}_{p_{pre}(x_0|z)}[-(r(x_0,z)+\gamma)\log p_\theta(x_0|z)]$$
$$\leqslant \mathbb{E}_{p(z)}\mathbb{E}_{p_{pre}(x_0|z)}[\sum_{t>1}\mathbb{E}_{q(x_t|x_0,z)}[\frac{r(x_0,z)||\tilde{\mu}_t(x_t,x_0)-\mu_\theta(x_t,t,z)||^2}{2\sigma_t^2}$$
$$+\frac{\gamma(||\mu_{pre}(x_t,t,z)-\mu_\theta(x_t,t,z)||^2)}{2\sigma_t^2}]] + C_2. \tag{12}$$

In Lemma 4.3, we introduce KL regularization (Eq. (11)) for supervised fine-tuning, which can be incorporated by adjusting the original reward with a shift factor $\gamma$ in the reward-weighted loss, smoothing the weighting of each sample towards the uniform distribution.[2] We refer to this regularization as KL-D since it is based only on data from the pre-trained model. The induced supervised training objective for KL-D is as follows:

$$\mathbb{E}_{p(z)}\mathbb{E}_{p_{\text{pre}}(x_0|z)}\left[(r(x_0,z)+\gamma)\sum_{t>1}\mathbb{E}_{q(x_t|x_0,z)}\left[\frac{||x_0-f_\theta(x_t,t,z)||^2}{2\sigma_t^2}\right]\right]. \tag{13}$$

We also consider another KL regularization presented in Eq. (12). This KL regularization can be implemented by introducing an additional term in the reward-weighted loss. This extra term penalizes the $L_2$-distance between the denoising directions derived from the pre-trained and current models. We refer to it as KL-O because it also regularizes the output from the current model to be close to that from the pre-trained model. The induced supervised training objective for KL-O is as follows:

$$\mathbb{E}_{p(z)}\mathbb{E}_{p_{\text{pre}}(x_0|z)}\left[\sum_{t>1}\mathbb{E}_{q(x_t|x_0,z)}\left[\frac{r(x_0,z)\|x_0-f_\theta(x_t,t,z)\|^2 + \gamma\|f_{\text{pre}}(x_t,t,z)-f_\theta(x_t,t,z)\|^2}{2\sigma_t^2}\right]\right].$$

We summarize our supervised fine-tuning in Algorithm 2. Note that in comparison to online RL training, our supervised setting only requires a pre-trained diffusion model and no extra/new datasets.

### 4.3 Online RL vs. Supervised Fine-tuning

We outline key differences between online RL and supervised fine-tuning:

- **Online versus offline distribution.** We first contrast their objectives. The online RL objective is to find a new image distribution that maximizes expected reward—this can have much different support than the pre-trained distribution. In supervised fine-tuning, the objective only encourages the model to "imitate" good examples from the supervised dataset, which always lies in the support of the pre-trained distribution.

---

[2]Intuitively, adjusting $\gamma$ is similar to adjusting the temperature parameter in reward-weighted regression [24].

---

**Algorithm 2** Supervised fine-tuning with KL regularization

---

**Input**: Reward model $r$, pre-trained diffusion model $p_{\text{pre}}$, diffusion model $p_\theta$, regularization weight $\gamma$, batch size $n$, the number of training samples $M$

    Initialize $p_\theta = p_{\text{pre}}$

    Collect $M$ samples from pre-trained model: $\mathcal{D} = \{(x_0, z) \sim p_{\text{pre}}(x_0|z)p(z)|\forall i \in \{1, ..., M\}\}$

    **while** $\theta$ not converged **do**

        Obtain $n$ i.i.d. samples $\{x_0, z\}$ from training dataset $\mathcal{D}$

        Sample $x_t$ given $x_0$, $t \sim [1, T]$

        For KL-D, compute the gradient $\nabla_\theta \frac{(r(x_0,z)+\gamma)||x_0-f_\theta(x_t,t,z)||^2}{2\sigma_t^2}$, update $\theta$

        For KL-O, compute the gradient $\nabla_\theta \frac{r(x_0,z)\|x_0-f_\theta(x_t,t,z)\|^2+\gamma\|f_{\text{pre}}(x_t,t,z)-f_\theta(x_t,t,z)\|^2}{2\sigma_t^2}$, update $\theta$

    **end while**

**Output**: Fine-tuned diffusion model $p_\theta$

---

- **Different KL regularization.** The methods also differ in their use of KL regularization. For online training, we evaluate conditional KL in each step using online samples, while for supervised training we evaluate conditional KL only using supervised samples. Moreover, online KL regularization can be seen as an extra reward function to encourage small divergence w.r.t. the pre-trained model for online optimization, while supervised KL induces an extra shift in the original reward for supervised training as shown in Lemma 4.3.

- **Different evaluation of the reward model.** Online RL fine-tuning evaluates the reward model using the updated distribution, while supervised fine-tuning evaluates the reward on the fixed pre-training data distribution. As a consequence, our online RL optimization should derive greater benefit from the generalization ability of the reward model.

For these reasons, we expect online RL fine-tuning and supervised fine-tuning to generate rather different behaviors. Specifically, online fine-tuning should be better at maximizing the combined reward (human reward and implicit KL reward) than the supervised approach (similar to the difference between online learning and weighted behavior cloning in reinforcement learning).

## 5 Experimental Evaluation

We now describe a set of experiments designed to test the efficacy of different fine-tuning methods.

### 5.1 Experimental Setup

As our baseline generative model, we use Stable Diffusion v1.5 [30], which has been pre-trained on large image-text datasets [33, 34]. For compute-efficient fine-tuning, we use Low-Rank Adaption (LoRA) [11], which freezes the parameters of the pre-trained model and introduces low-rank trainable weights. We apply LoRA to the UNet [31] module and only update the added weights. For the reward model, we use ImageReward [43] which is trained on a large dataset comprised of human assessments of images. Compared to other scoring functions such as CLIP [27] or BLIP [18], ImageReward has a better correlation with human judgments, making it the preferred choice for fine-tuning our baseline diffusion model (see Appendix C for further justification). Further experimental details (e.g., model architectures, final hyper-parameters) are provided in Appendix B. We also provide more samples for qualitative comparison in Appendix E.6.

### 5.2 Comparison of Supervised and RL Fine-tuning

We first evaluate the performance of the original and fine-tuned text-to-image models w.r.t. specific capabilities, such as generating objects with specific colors, counts, or locations; and composing multiple objects (or *composition*). For both RL and SFT training we include KL regularization (see hyperparameters in Appendix B). For SFT, we adopt KL-O since we found it is more effective than KL-D in improving the visual quality of the SFT model (see the comparison between KL-O and KL-D in Appendix E.1). To systematically analyze the effects of different fine-tuning methods, we adopt a straightforward setup that uses one text prompt during fine-tuning. As training text prompts,

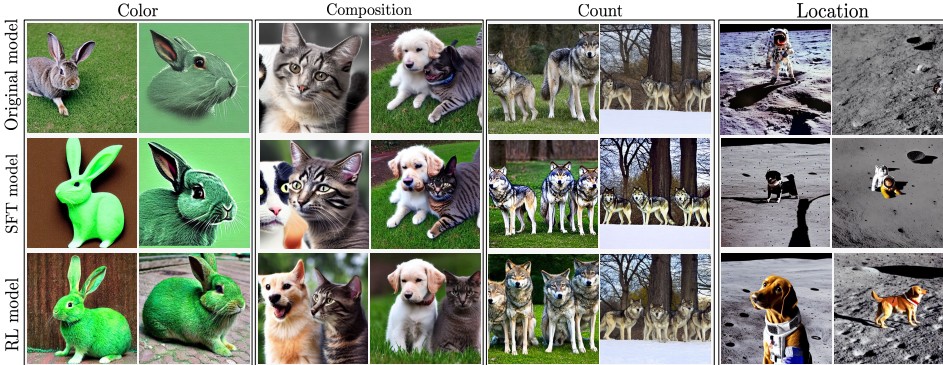

Figure 2: Comparison of images generated by the original Stable Diffusion model, supervised fine-tuned (SFT) model, and RL fine-tuned model. Images in the same column are generated with the same random seed. Images from seen text prompts: "A green colored rabbit" (color), "A cat and a dog" (composition), "Four wolves in the park" (count), and "A dog on the moon" (location).

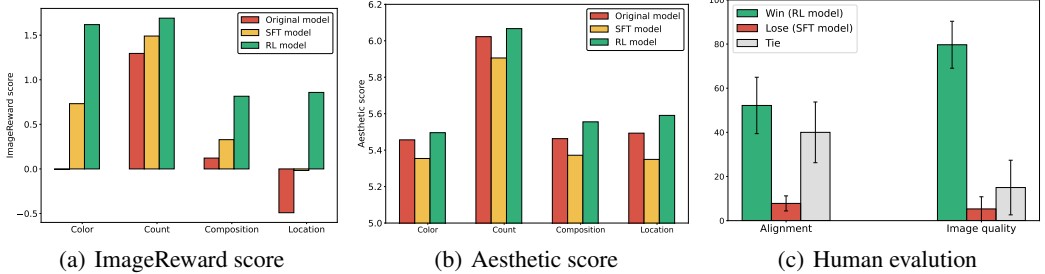

(a) ImageReward score      (b) Aesthetic score      (c) Human evalution

Figure 3: (a) ImageReward scores and (b) Aesthetic scores of three models: the original model, supervised fine-tuned (SFT) model, and RL fine-tuned model. ImageReward and Aesthetic scores are averaged over 50 samples from each model. (c) Human preference rates between RL model and SFT model in terms for image-text alignment and image quality. The results show the mean and standard deviation averaged over eight independent human raters.

we use "A green colored rabbit", "A cat and a dog", "Four wolves in the park", and "A dog on the moon". These test the models' ability to handle prompts involving *color*, *composition*, *counting* and *location*, respectively. For supervised fine-tuning, we use 20K images generated by the original model, which is the same number of (online) images used by RL fine-tuning.

Figure 3(a) compares ImageReward scores of images generated by the different models (with the same random seed). We see that both SFT and RL fine-tuning improve the ImageReward scores on the training text prompt. This implies the fine-tuned models can generate images that are better aligned with the input text prompts than the original model because ImageReward is trained on human feedback datasets to evaluate image-text alignment. Figure 2 indeed shows that fine-tuned models add objects to match the number (e.g., adding more wolves in "Four wolves in the park"), and replace incorrect objects with target objects (e.g., replacing an astronaut with a dog in "A dog on the moon") compared to images from the original model. They also avoid obvious mistakes like generating a rabbit with a green background given the prompt "A green colored rabbit". Of special note, we find that the fine-tuned models generate better images than the original model on several unseen text prompts consisting of unseen objects in terms of image-text alignment (see Figure 10 in Appendix E).

As we can observe in Figure 3(a), RL models enjoy higher ImageReward than SFT models when evaluated with the same training prompt in different categories respectively, due to the benefit of online training as we discuss in Section 4.3. We also evaluate the image quality of the three models using the aesthetic predictor [34], which is trained to predict the aesthetic aspects of generated images.[3] Figure 3(b) shows that supervised fine-tuning degrades image quality relative to the RL approach (and sometimes relative to the pre-trained model), even with KL regularization which is

---

[3]The aesthetic predictor had been utilized to filter out the low-quality images in training data for Stable Diffusion models.

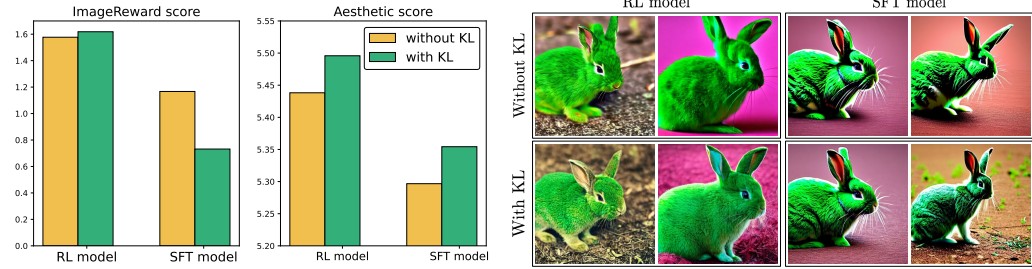

(a) ImageReward and Aesthetic scores      (b) Generated images

Figure 4: Ablation study of KL regularization in both SFT and RL training, trained on a single prompt "A green colored rabbit". (a) ImageReward and Aesthetic scores are averaged over 50 samples from each model. (b) Images generated by RL models and SFT models optimized with and without KL regularization. Images in the same column are generated with the same random seed.

Original model          Fine-tuned model

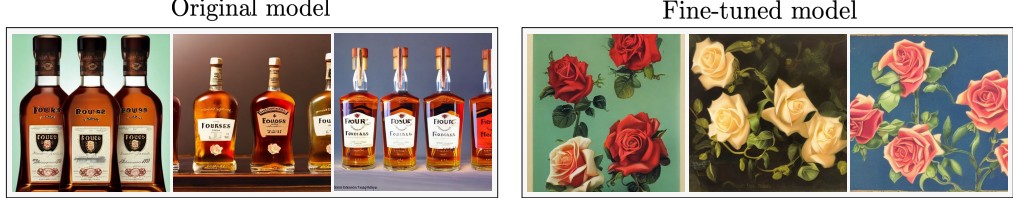

Figure 5: Comparison of images generated by the original model and RL fine-tuned model on text prompt "Four roses". The original model, which is trained on large-scale datasets from the web [34, 33], tends to produce whiskey-related images from "Four roses" due to the existence of a whiskey brand bearing the same name as the prompt. In contrast, RL fine-tuned model with ImageReward generates images associated with the flower "rose".

intended to blunt such effects. For example, the supervised model often generates over-saturated images, corroborating the observations of Lee et al. [17]. By contrast, the RL model generates more natural images that are at least as well-aligned with text prompts.

We also conduct human evaluation as follows: We collect 40 images randomly generated from each prompt, resulting in a total of 160 images for each model. Given two (anonymized) sets of four images from the same random seeds, one from RL fine-tuned model and one from the SFT model, we ask human raters to assess which one is better w.r.t. image-text alignment and image quality. Each query is evaluated by 8 independent raters and we report the average win/lose rate in Figure 3(c). The RL model consistently outperforms the SFT model on both alignment and image quality.

### 5.3 The Effect of KL Regularization

To demonstrate the impact of KL regularization on both supervised fine-tuning (SFT) and online RL fine-tuning, we conduct an ablation study specifically fine-tuning the pre-trained model with and without KL regularization on "A green colored rabbit".

Figure 4(a) shows that KL regularization in online RL is effective in attaining both high reward and aesthetic scores. We observe that the RL model without KL regularization can generate lower-quality images (e.g., over-saturated colors and unnatural shapes) as shown in Figure 4(b). In the case of SFT with KL-O (where we use the same configuration as Section 5.2), we find that the KL regularization can mitigate some failure modes of SFT without KL and improve aesthetic scores, but generally suffers from lower ImageReward. We expect that this difference in the impact of KL regularization is due to the different nature of online and offline training—KL regularization is only applied to fixed samples in the case of SFT while KL regularization is applied to new samples per each update in the case of online RL (see Section 4.3 for related discussions).

### 5.4 Reducing Bias in the Pre-trained Model

To see the benefits of optimizing for ImageReward, we explore the effect of RL fine-tuning in reducing bias in the pre-trained model. Because the original Stable Diffusion model is trained on

large-scale datasets extracted from the web [33, 34], it can encode some biases in the training data. As shown in Figure 5, we see that the original model tends to produce whiskey-related images given the prompt "Four roses" (which happens to be the brand name of a whiskey), which is not aligned with users' intention in general. To verify whether maximizing a reward model derived from human feedback can mitigate this issue, we fine-tune the original model on the "Four roses" prompt using RL fine-tuning. Figure 5 shows that the fine-tuned model generates images with roses (the flower) because the ImageReward score is low on the biased images (ImageReward score is increased to 1.12 from -0.52 after fine-tuning). This shows the clear benefits of learning from human feedback to better align existing text-to-image models with human intentions.

## 5.5 Fine-tuning on Multiple Prompts

We further verify the effectiveness of the proposed techniques for fine-tuning text-to-image models on multiple prompts simultaneously. We conduct online RL training with 104 MS-CoCo prompts and 183 Drawbench prompts, respectively (the prompts are randomly sampled during training). Detailed configurations are provided in Appendix E. Specifically, we also learn a value function for variance reduction in policy gradient which shows benefit in further improving the final reward (see Appendix A.5 for details.) We report both ImageReward and the aesthetic score of the original and the RL fine-tuned models. For evaluation, we generate 30 images from each prompt and report the average scores of all images. The evaluation result is reported in Table 1 with sample images in Figure 11 in Appendix E, showing that RL training can also significantly improve the ImageReward score while maintaining a high aesthetic score with much larger sets of training prompts.

| | MS-CoCo | | Drawbench | |
| --- | --- | --- | --- | --- |
| | Original model | RL model | Original model | RL model |
| ImageReward score | 0.22 | 0.55 | 0.13 | 0.58 |
| Aesthetic score | 5.39 | 5.43 | 5.31 | 5.35 |

Table 1: ImageReward scores and Aesthetic scores from the original model, and RL fine-tuned model on multiple prompts from MS-CoCo (104 prompts) and Drawbench (183 prompts). We report the average ImageReward and Aesthetic scores across 3120 and 5490 images on MS-CoCo and Drawbench, respectively (30 images per each prompt).

## 6 Discussions

In this work, we propose DPOK, an algorithm to fine-tune a text-to-image diffusion model using policy gradient with KL regularization. We show that online RL fine-tuning outperforms simple supervised fine-tuning in improving the model's performance. Also, we conduct an analysis of KL regularization for both methods and discuss the key differences between RL fine-tuning and supervised fine-tuning. We believe our work demonstrates the potential of *reinforcement learning from human feedback* in improving text-to-image diffusion models.

**Limitations, future directions.** Several limitations of our work suggest interesting future directions: (a) As discussed in Section 5.5, fine-tuning on multiple prompts requires longer training time, hyper-parameter tuning, and engineering efforts. More efficient training with a broader range of diverse and complex prompts would be an interesting future direction to explore. (b) Exploring advanced policy gradient methods can be useful for further performance improvement. Investigating the applicability and benefits of these methods could lead to improvements in the fine-tuning process.

**Broader Impacts** Text-to-image models can offer societal benefits across fields such as art and entertainment, but they also carry the potential for misuse. Our research allows users to finetune these models towards arbitrary reward functions. This process can be socially beneficial or harmful depending on the reward function used. Users could train models to produce less biased or offensive imagery, but they could also train them to produce misinformation or deep fakes. Since many users may follow our approach of using open-sourced reward functions for this finetuning, it is critical that the biases and failure modes of publicly available reward models are thoroughly documented.

## Acknowledgements

We thank Deepak Ramachandran and Jason Baldridge for providing helpful comments and suggestions. Support for this research was also provided by the University of Wisconsin-Madison Office of the Vice Chancellor for Research and Graduate Education, NSF Award DMS-2023239, NSF/Intel Partnership on Machine Learning for Wireless Networking Program under Grant No. CNS-2003129, and FuriosaAI. OW and YD are funded by the Center for Human Compatible Artificial Intelligence.

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

# Appendix:

# Reinforcement Learning for Fine-tuning Text-to-Image Diffusion Models

## A Derivations

### A.1 Lemma 4.1

*Proof.*

$$\nabla_\theta \mathbb{E}_{p(z)} \mathbb{E}_{p_\theta(x_0|z)} [-r(x_0, z)] = -\mathbb{E}_{p(z)} \left[ \nabla_\theta \int p_\theta(x_0|z) r(x_0, z) dx_0 \right]$$

$$= -\mathbb{E}_{p(z)} \left[ \nabla_\theta \int \left( \int p_\theta(x_{0:T}|z) dx_{1:T} \right) r(x_0, z) dx_0 \right]$$

$$= -\mathbb{E}_{p(z)} \left[ \int \nabla_\theta \log p_\theta(x_{0:T}|z) \times r(x_0, z) \times p_\theta(x_{0:T}|z) \ dx_{0:T} \right]$$

$$= -\mathbb{E}_{p(z)} \left[ \int \nabla_\theta \log \left( p_T(x_T|z) \prod_{t=1}^T p_\theta(x_{t-1}|x_t, z) \right) \times r(x_0, z) \times p_\theta(x_{0:T}|z) \ dx_{0:T} \right]$$

$$= -\mathbb{E}_{p(z)} \mathbb{E}_{p_\theta(x_{0:T}|z)} \left[ r(x_0, z) \sum_{t=1}^T \nabla_\theta \log p_\theta(x_{t-1}|x_t, z) \right],$$

where the second to last equality is from the continuous assumptions of $p_\theta(x_{0:T}|z) r(x_0, z)$ and $\nabla_\theta p_\theta(x_{0:T}|z) r(x_0, z)$. □

### A.2 Lemma 4.2

Similar to the proof in Theorem 1 in [40], from data processing inequality with the Markov kernel, given any $z$ we have

$$\text{KL}(p_\theta(x_0|z) \| p_{\text{pre}}(x_0|z)) \leqslant \text{KL}(p_\theta(x_{0:T}|z) \| p_{\text{pre}}(x_{0:T}|z)). \tag{14}$$

Using the Markov property of $p_\theta$ and $p_{\text{pre}}$, we have

$$\text{KL}(p_\theta(x_{0:T}|z) \| p_{\text{pre}}(x_{0:T}|z)) = \int p_\theta(x_{0:T}|z) \times \log \frac{p_\theta(x_{0:T}|z)}{p_{\text{pre}}(x_{0:T}|z)} \ dx_{0:T}$$

$$= \int p_\theta(x_{0:T}|z) \log \frac{p_\theta(x_T|z) \prod_{t=1}^T p_\theta(x_{t-1}|x_t, z)}{p_{\text{pre}}(x_T|z) \prod_{t=1}^T p_{\text{pre}}(x_{t-1}|x_t, z)} dx_{0:T}$$

$$= \int p_\theta(x_{0:T}|z) \left( \log \frac{p_\theta(x_T|z)}{p_{\text{pre}}(x_T|z)} + \sum_{t=1}^T \log \frac{p_\theta(x_{t-1}|x_t, z)}{p_{\text{pre}}(x_{t-1}|x_t, z)} \right) dx_{0:T}$$

$$= \mathbb{E}_{p_\theta(x_{0:T}|z)} \left[ \sum_{t=1}^T \log \frac{p_\theta(x_{t-1}|x_t, z)}{p_{\text{pre}}(x_{t-1}|x_t, z)} \right] = \sum_{t=1}^T \mathbb{E}_{p_\theta(x_{t:T}|z)} \mathbb{E}_{p_\theta(x_{0:t-1}|x_{t:T}, z)} \left[ \log \frac{p_\theta(x_{t-1}|x_t, z)}{p_{\text{pre}}(x_{t-1}|x_t, z)} \right]$$

$$= \sum_{t=1}^T \mathbb{E}_{p_\theta(x_t|z)} \mathbb{E}_{p_\theta(x_{0:t-1}|x_t, z)} \left[ \log \frac{p_\theta(x_{t-1}|x_t, z)}{p_{\text{pre}}(x_{t-1}|x_t, z)} \right] = \sum_{t=1}^T \mathbb{E}_{p_\theta(x_t|z)} \mathbb{E}_{p_\theta(x_{t-1}|x_t, z)} \left[ \log \frac{p_\theta(x_{t-1}|x_t, z)}{p_{\text{pre}}(x_{t-1}|x_t, z)} \right]$$

$$= \sum_{t=1}^T \mathbb{E}_{p_\theta(x_t|z)} \big[ \text{KL} \big( p_\theta(x_{t-1}|x_t, z) \| p_{\text{pre}}(x_{t-1}|x_t, z) \big) \big].$$

Taking the expectation of $p(z)$ on both sides, we have

$$\mathbb{E}_{p(z)}[\mathrm{KL}(p_\theta(x_0|z))\|p_{\mathrm{pre}}(x_0|z))] \leqslant \mathbb{E}_{p(z)}\left[\sum_{t=1}^{T}\mathbb{E}_{p_\theta(x_t|z)}\big[\mathrm{KL}\big(p_\theta(x_{t-1}|x_t,z)\|p_{\mathrm{pre}}(x_{t-1}|x_t,z)\big)\big]\right].$$
(15)

### A.3 The gradient of objective Eq. (8)

From Lemma 4.2, for online fine-tuning, we need to regularize $\sum_{t=1}^{T}\mathbb{E}_{p_\theta(x_t)}\big[\mathrm{KL}\big(p_\theta(x_{t-1}|x_t,z)\|p_{\mathrm{pre}}(x_{t-1}|x_t,z)\big)\big]$. By the product rule, we have

$$\nabla_\theta \sum_{t=1}^{T}\mathbb{E}_{p_\theta(x_t)}\big[\mathrm{KL}\big(p_\theta(x_{t-1}|x_t,z)\|p_{\mathrm{pre}}(x_{t-1}|x_t,z)\big)\big]$$

$$= \sum_{t=1}^{T}\mathbb{E}_{p_\theta(x_t)}\big[\nabla_\theta\mathrm{KL}\big(p_\theta(x_{t-1}|x_t,z)\|p_{\mathrm{pre}}(x_{t-1}|x_t,z)\big)\big] \quad (16)$$

$$+ \sum_{t=1}^{T}\mathbb{E}_{p_\theta(x_t)}\big[\sum_{t'>t}\nabla_\theta\log p_\theta(x_{t'-1}|x_{t'},z)\cdot\mathrm{KL}\big(p_\theta(x_{t-1}|x_t,z)\|p_{\mathrm{pre}}(x_{t-1}|x_t,z)\big)\big],$$

which treats the sum of conditional KL-divergences along the future trajectory as a scalar reward at each step. However, computing these sums is more inefficient than just the first term in Eq. (16). Empirically, we find that regularizing only the first term in Eq. (16) already works well, so we adopt this approach for simplicity.

### A.4 Lemma 4.3

Similar to derivation in [10], and notice that $q$ is not dependent on $z$ (i.e., $q(\cdots|z) = q(\cdots)$), we have

$$\mathbb{E}_{p(z)}\mathbb{E}_{p_{\mathrm{pre}}(x_0|z)}[-(r(x_0,z)+\gamma)\log p_\theta(x_0|z)]$$

$$\leqslant \mathbb{E}_{p(z)}\mathbb{E}_{p_{\mathrm{pre}}(x_0|z)}\mathbb{E}_{q(x_{1:T}|x_0,z)}\left[-(r(x_0,z)+\gamma)\log\frac{p_\theta(x_{0:T}|z)}{q(x_{1:T}|x_0,z)}\right]$$

$$= \mathbb{E}_{p(z)}\mathbb{E}_{p_{\mathrm{pre}}(x_0|z)}\mathbb{E}_{q(x_{1:T}|x_0,z)}[-(r(x_0,z)+\gamma)\log\frac{p(x_T|z)}{q(x_T|x_0,z)}$$

$$-\sum_{t>1}(r(x_0,z)+\gamma)\log\frac{p_\theta(x_{t-1}|x_t,z)}{q(x_{t-1}|x_t,x_0,z)} \quad (17)$$

$$-(r(x_0,z)+\gamma)\log p_\theta(x_0|x_1,z)]$$

$$= \mathbb{E}_{p(z)}\mathbb{E}_{p_{\mathrm{pre}}(x_0|z)}[(r(x_0,z)+\gamma)\mathbb{E}_{q(x_T|x_0,z)}[\mathrm{KL}(q(x_T|x_0)\|p(x_T))]$$

$$+(r(x_0,z)+\gamma)\sum_{t>1}\mathbb{E}_{q(x_t|x_0,z)}[\mathrm{KL}(q(x_{t-1}|x_t,x_0)\|p_\theta(x_{t-1}|x_t,z))]$$

$$-(r(x_0,z)+\gamma)\mathbb{E}_{q(x_1|x_0,z)}[\log p_\theta(x_0|x_1,z)]],$$

where the first inequality comes from ELBO.

Let $\tilde{\mu}_t(x_t,x_0) := \frac{\sqrt{\bar{\alpha}_{t-1}}\beta_t}{1-\bar{\alpha}_t}x_0 + \frac{\sqrt{\alpha_t}(1-\bar{\alpha}_{t-1})}{1-\bar{\alpha}_t}x_t$ and $\tilde{\beta}_t = \frac{1-\bar{\alpha}_{t-1}}{1-\bar{\alpha}_t}\beta_t$, we have:

$$\mathbb{E}_{p(z)}\mathbb{E}_{p_{\mathrm{pre}}(x_0|z)}[(r(x_0,z)+\gamma)\sum_{t>1}\mathbb{E}_{q(x_t|x_0,z)}[\mathrm{KL}(q(x_{t-1}|x_t,x_0)\|p_\theta(x_{t-1}|x_t,z))]]$$

$$= \mathbb{E}_{p(z)}\mathbb{E}_{p_{\mathrm{pre}}(x_0|z)}[(r(x_0,z)+\gamma)\sum_{t>1}\mathbb{E}_{q(x_t|x_0,z)}[\frac{1}{2\sigma_t^2}||\tilde{\mu}_t(x_t,x_0)-\mu_\theta(x_t,t,z)||^2]] + C, \quad (18)$$

where $C$ is a constant.

Notice that $p_\theta(x_0|z,x_1)$ is modeled as a discrete decoder as in [10] and $p(x_T|z)$ is a fixed Gaussian; neither is trainable.

As a result, we have

$$\mathbb{E}_{p(z)}\mathbb{E}_{p_{\mathrm{pre}}(x_0|z)}[-(r(x_0,z)+\gamma)\log p_\theta(x_0|z)]$$

$$\leqslant \mathbb{E}_{p(z)}\mathbb{E}_{p_{\mathrm{pre}}(x_0|z)}[(r(x_0,z)+\gamma)\sum_{t>1}\mathbb{E}_{q(x_t|x_0,z)}[\frac{1}{2\sigma_t^2}||\tilde{\mu}_t(x_t,x_0)-\mu_\theta(x_t,t,z)||^2]] + C_1 \quad (19)$$

Furthermore, from triangle inequality we have

$$\sum_{t>1}\mathbb{E}_{q(x_t|x_0,z)}[\gamma\frac{||\tilde{\mu}_t(x_t,x_0)-\mu_\theta(x_t,t,z)||^2}{2\sigma_t^2}]$$

$$\leqslant \sum_{t>1}\mathbb{E}_{q(x_t|x_0,z)}[\gamma\frac{||\tilde{\mu}_t(x_t,x_0)-\mu_{\mathrm{pre}}(x_t,t,z)||^2 + ||\mu_{\mathrm{pre}}(x_t,t,z)-\mu_\theta(x_t,t,z)||^2}{2\sigma_t^2}] \quad (20)$$

So another weaker upper bound is given by

$$\mathbb{E}_{p(z)}\mathbb{E}_{p_{\mathrm{pre}}(x_0|z)}[-(r(x_0,z)+\gamma)\log p_\theta(x_0|z)]$$

$$\leqslant \mathbb{E}_{p(z)}\mathbb{E}_{p_{\mathrm{pre}}(x_0|z)}[\sum_{t>1}\mathbb{E}_{q(x_t|x_0,z)}[\frac{r(x_0,z)||\tilde{\mu}_t(x_t,x_0)-\mu_\theta(x_t,t,z)||^2}{2\sigma_t^2}$$

$$+\frac{\gamma(||\mu_{\mathrm{pre}}(x_t,t,z)-\mu_\theta(x_t,t,z)||^2)}{2\sigma_t^2}]] + C_2. \quad (21)$$

Notice that $p_\theta(x_0|z,x_1)$ is modeled as a discrete decoder as in [10] and $p(x_T|z)$ is a fixed Gaussian; neither is trainable. Then the proof is complete.

## A.5  Value function learning

Similar to [35], we can also apply the variance reduction trick by subtracting a baseline function $V(x_t,z,\theta)$:

$$\mathbb{E}_{p(z)}\mathbb{E}_{p_\theta(x_{0:T}|z)}\left[-r(x_0,z)\sum_{t=1}^T \nabla_\theta \log p_\theta(x_{t-1}|x_t,z)\right]$$

$$= \mathbb{E}_{p(z)}\mathbb{E}_{p_\theta(x_{0:T}|z)}\left[-\sum_{t=1}^T (r(x_0,z)-V(x_t,z,\theta))\nabla_\theta \log p_\theta(x_{t-1}|x_t,z)\right], \quad (22)$$

where

$$V(x_t,z,\theta) := \mathbb{E}_{p_\theta(x_{0:t})}[r(x_0,z)|x_t]. \quad (23)$$

So we can learn a value function estimator by minimizing the objective function below for each $t$:

$$\mathbb{E}_{p_\theta(x_{0:t})}\left[\left(r(x_0,z)-\hat{V}(x_t,\theta,z)\right)^2|x_t\right]. \quad (24)$$

Since subtracting $V(x_t,\theta,z)$ will minimize the variance of the gradient estimation, it is expected that such a trick can improve policy gradient training. In our experiments, we find that it can also improve the final reward: For example, for the prompt "A dog on the moon", adding variance reduction can improve ImageReward from 0.86 to 1.51, and also slightly improve the aesthetic score from 5.57 to 5.60. The generated images with and without value learning are shown in Figure 6. We also find similar improvements in multi-prompt training (see learning curves with and without value learning in Figure 7).

| With value learning | Without value learning |
|---|---|
| 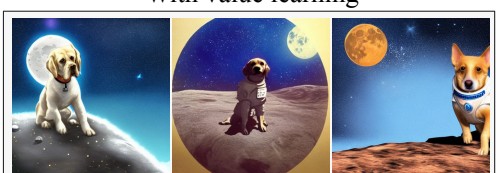 | 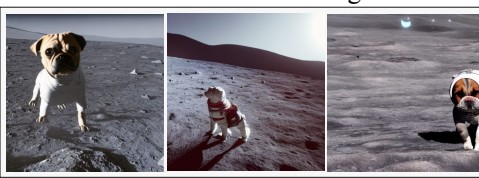 |

Figure 6: Text prompt: "A dog on the moon", generated from the same random seeds from the model trained with and without value learning, respectively.

## A.6 Importance sampling and ratio clipping

In order to reuse old trajectory samples, we can apply the important sampling trick:

$$
\begin{aligned}
&\mathbb{E}_{p(z)}\mathbb{E}_{p_\theta(x_{0:T}|z)}\left[-\sum_{t=1}^{T}\left(r(x_0,z)-V(x_t,z,\theta)\right)\nabla_\theta\log p_\theta(x_{t-1}|x_t,z)\right]\\
&=\mathbb{E}_{p(z)}\mathbb{E}_{p_{\theta_{old}}(x_{0:T}|z)}\left[-\sum_{t=1}^{T}\left(r(x_0,z)-V(x_t,z,\theta)\right)\frac{p_\theta(x_{t-1}|x_t,z)}{p_{\theta_{old}}(x_{t-1}|x_t,z)}\nabla_\theta\log p_\theta(x_{t-1}|x_t,z)\right]\\
&=\mathbb{E}_{p(z)}\mathbb{E}_{p_{\theta_{old}}(x_{0:T}|z)}\left[-\sum_{t=1}^{T}\left(r(x_0,z)-V(x_t,z,\theta)\right)\nabla_\theta\frac{p_\theta(x_{t-1}|x_t,z)}{p_{\theta_{old}}(x_{t-1}|x_t,z)}\right].
\end{aligned}
\tag{25}
$$

In order to constrain $p_\theta$ to be close to $p_{\theta_{old}}$, we can use the clipped gradient for policy gradient update similar to PPO [36]:

$$
\mathbb{E}_{p(z)}\mathbb{E}_{p_{\theta_{old}}(x_{0:T}|z)}\left[-\sum_{t=1}^{T}\left(r(x_0,z)-V(x_t,z,\theta)\right)\nabla_\theta\mathrm{clip}\left(\frac{p_\theta(x_{t-1}|x_t,z)}{p_{\theta_{old}}(x_{t-1}|x_t,z)},1+\epsilon,1-\epsilon\right)\right],
\tag{26}
$$

where $\epsilon$ is the clip hyperparameter.

## B Experimental Details

**Online RL training.** For hyper-parameters of online RL training used in Section 5.2., we use $\alpha=10, \beta=0.01$, learning rate = $10^{-5}$ and keep other default hyper-parameters in AdamW, sampling batch size $m=10$. For policy gradient training, to perform more gradient steps without the need of re-sampling, we perform 5 gradient steps per sampling step, with gradient norm clipped to be smaller than 0.1, and use importance sampling to handle off-policy samples, and use batch size $n=32$ in each gradient step. We train the models till generating 20000 online samples, which matches the number of samples in supervised fine-tuning and results in 10K gradient steps in total. For stable training, we freeze the weights in batch norm to be exactly the same as the pretrained diffusion model. For importance sampling, we apply the clip hyperparameter $\epsilon=10^{-4}$.

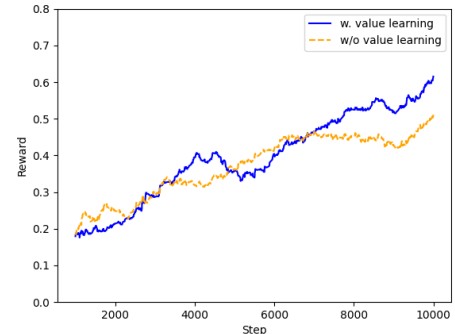

Figure 7: Learning curves with and without value learning, trained on the Drawbench prompt set: Adding value learning could result in higher reward using less time.

**Supervised training.** For hyper-parameters of supervised training, we use $\gamma=2.0$ as the default option in Section 5.2, which is chosen from $\gamma\in\{0.1,1.0,2.0,5.0\}$. We use learning rate $=2\times10^{-5}$ and keep other default hyper-parameters in AdamW, which was chosen from $\{5\times10^{-6},1\times10^{-5},2\times10^{-5},5\times10^{-5}\}$. We use batch size

$n = 128$ and $M = 20000$ such that both algorithms use the same number of samples, and train the SFT model for 8K gradient steps. Notice that Lemma 4.3 requires $r + \gamma$ to be non-negative, we apply a filter of reward such that if $r + \gamma < 0$, we just set it to 0. In practice, we also observe that without this filtering, supervised fine-tuning could fail during training, which indicates that our assumption is indeed necessary.

## C Investigation on ImageReward

We investigate the quality of ImageReward [43] by evaluating its prediction of the assessments of human labelers. Similar to Lee et al. [17], we check whether ImageReward generates a higher score for images preferred by humans. Our evaluation encompasses four text prompt categories: color, count, location, and composition. To generate prompts, we combine words or phrases from each category with various objects. These prompts are then used to generate corresponding images using a pre-trained text-to-image model. We collect binary feedback from human labelers on the image-text dataset and construct comparison pairs based on this feedback. Specifically, we utilize 804, 691, 1154 and 13228 comparison pairs obtained from 38, 29, 41 and 295 prompts for color, location, count, and composition, respectively.[4] Additionally, for evaluation on complex text prompts from human users, we also utilize the test set from ImageReward, which consists of 6399 comparison pairs from 466 prompts.[5] Table 2 compares the accuracy of ImageReward and baselines like CLIP [27] and BLIP [18] scores, which justifies the choice of using ImageReward for fine-tuning text-to-image models.

|  | Color | Count | Location | Composition | Complex |
|---|---|---|---|---|---|
| CLIP | 82.46 | 66.11 | **80.60** | 70.98 | 54.83 |
| BLIP | 77.36 | 61.61 | 78.14 | 70.98 | 57.79 |
| ImageReward | **86.06** | **73.65** | 79.73 | **79.65** | **65.13** |

Table 2: Accuracy (%) of CLIP score, BLIP score and ImageReward when predicting the assessments of human labelers.

## D A Comprehensive Discussion of Related Work

We summarize several similarities and differences between our work and a concurrent work [2] as follows:

- Both Black et al. [2] and our work explored online RL fine-tuning for improving text-to-image diffusion models. However, in our work, we provide theoretical insights for optimizing the reward with policy gradient methods and provide conditions for the equivalence to hold.

- Black et al. [2] demonstrated that RL fine-tuning can outperform supervised fine-tuning with reward-weighted loss [17] in optimizing the reward, which aligns with our own observations.

- In our work, we do not only focus on reward optimization: inspired by failure cases (e.g., over-saturated or non–photorealistic images) in supervised fine-tuning [17], we aim at finding an RL solution with KL regularization to solve the problem.

- Unique to our work, we systematically analyze KL regularization for both supervised and online fine-tuning with theoretical justifications. We show that KL regularization would be more effective in the online RL fine-tuning rather than the supervised fine-tuning. By adopting online KL regularization, our algorithm successfully achieves high rewards and maintains image quality without over-optimization issue.

---

[4]Full list of text prompts is available at [link].
[5]https://github.com/THUDM/ImageReward/blob/main/data/test.json

# E More Experimental Results

## E.1 KL-D vs KL-O: Ablation Study on Supervised KL Regularization

We also present the effects of both KL-D and KL-O with coefficient $\gamma \in \{0.1, 1.0, 2.0, 5.0\}$ in Figure 8 and Figure 9. We observe the followings:

- KL-D (eq. (11)) can slightly increase the aesthetic score. However, even with a large value of $\gamma$, it does not yield a significant improvement in visual quality. Moreover, it tends to result in lower ImageReward scores overall.

- In Figure 9, it is evident that KL-O is more noticeably effective in addressing the failure modes in supervised fine-tuning (SFT) compared to KL-D, especially when $\gamma$ is relatively large. However, as a tradeoff, KL-O significantly reduces the ImageReward scores in that case.

- In general, achieving both high ImageReward and aesthetic scores simultaneously is challenging for supervised fine-tuning (SFT), unlike RL fine-tuning (blue bar in Figure 8).

We also present more qualitative examples from different choices of KL regularization in Appendix E.5.

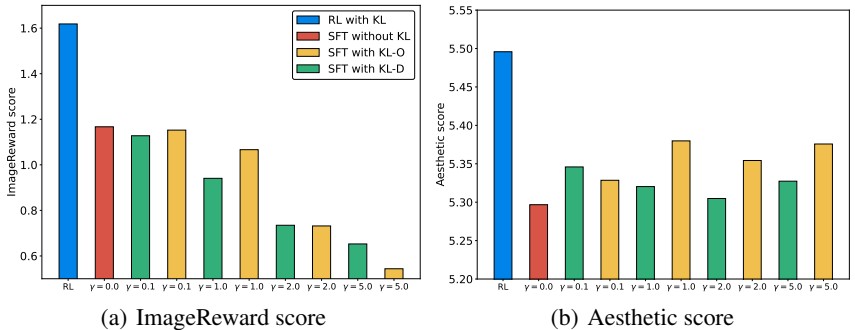

| (a) ImageReward score | (b) Aesthetic score |

Figure 8: (a) ImageReward scores and (b) Aesthetic scores of supervised fine-tuned (SFT) models with different choices of regularization term and coefficient ($\gamma$) on text prompt "A green colored rabbit". Each score is averaged over 50 samples from each model. KL-O and KL-D refer to eq. (12) and eq. (11), respectively.

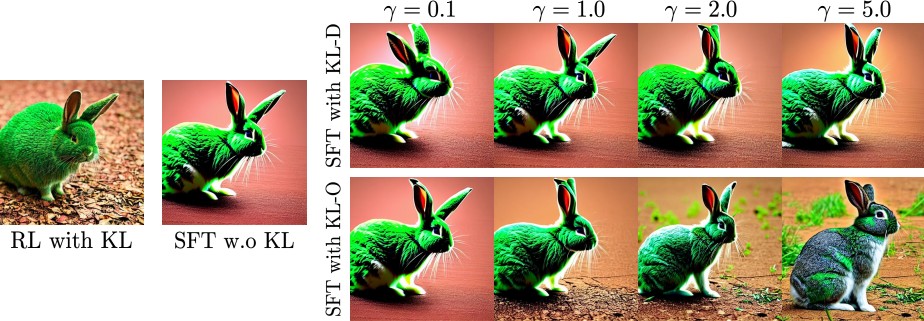

Figure 9: **(Left)** Sample from the RL model with KL regularization and sample from the SFT model without KL regularization. **(Right)** Samples from supervised fine-tuned models with different choices of regularization term and coefficient ($\gamma$) on "A green colored rabbit". As we increase $\gamma$ in the SFT case, there is a tradeoff between alignment with the prompt and image quality (i.e. oversaturation). On the other hand, the RL with KL sample is able to retain both alignment and image quality.

## E.2 Images from unseen text prompts

Figure 10 shows image samples from the original model, SFT model and RL fine-tuned models on unseen text prompts.

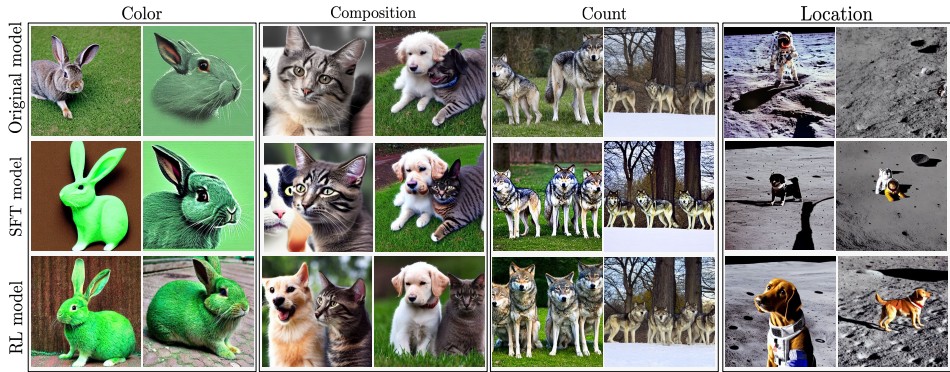

Figure 10: Comparison of images generated by the original Stable Diffusion model, supervised fine-tuned (SFT) model, and RL fine-tuned model. Images in the same column are generated with the same random seed. Images from unseen text prompts: "A green colored cat" (color), "A cat and a cup" (composition), "Four birds in the park" (count), and "A lion on the moon" (location).

## E.3 Multi-prompt Training

Here we report more details for multi-prompt training. We still use the same setting in Appendix B, but 1) increase the batch size per sampling step to $n = 45$ for stable training; 2) use a smaller KL weight $\beta = 0.001$; 3) use extra value learning for variance reduction. For value learning, we use a larger learning rate $10^{-4}$ and batch size 256. Also, we train for a longer time for multi-prompt such that it utilizes 50000 online samples. See sample images in Figure 11.

## E.4 Long and Complex Prompts

We show that our method does not just work for short and simple prompts like "A green colored rabbit", but can also work with long and complex prompts. For example, in Fig. 12 we adopt a complex prompt and compare the images generated by the original model and the supervised fine-tuned model. We can observe that online training encourages the models to generate images with a different style of painting and more fine-grained details, which is generally hard to achieve by supervised fine-tuning only (with KL-O regularization, $\gamma = 2.0$).

## E.5 More Qualitative Results from Ablation Study for KL Regularization in SFT

Here we provide samples from more prompts in KL ablation for SFT: see Figure 13.

## E.6 Qualitative Comparison

Here we provide more samples from the experiments in Section 5.2 and Section 5.3: see Figure 14, Figure 16, Figure 15, Figure 17, Figure 18 and Figure 19, with the same configuration as in Section 5.2 for both RL and SFT.

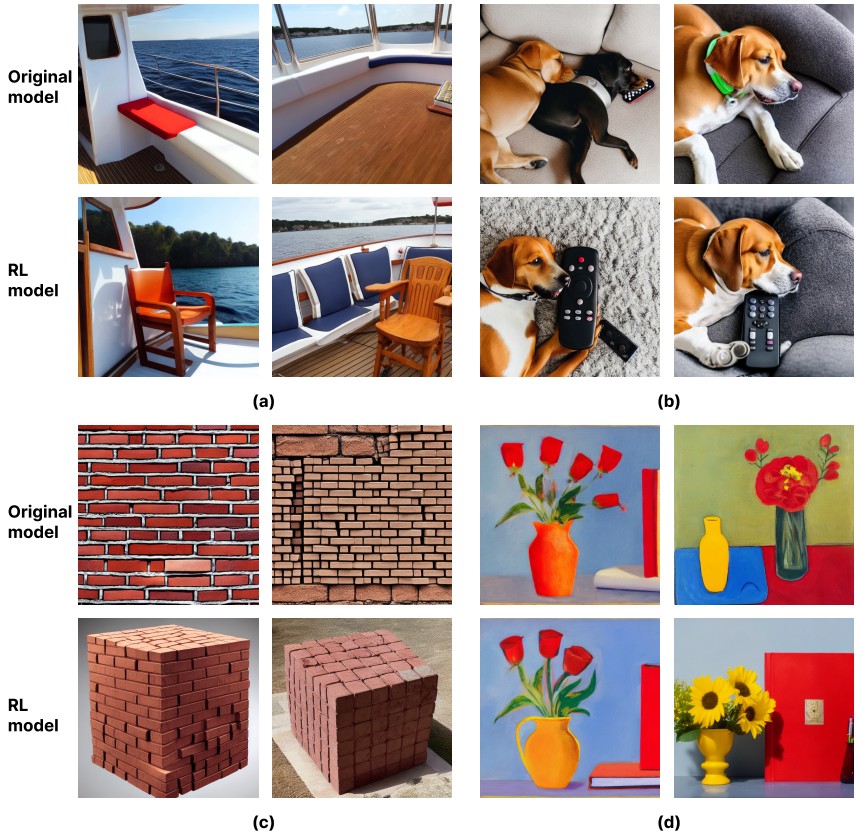

Figure 11: Sample images generated from prompts: (a) "A chair in the corner on a boat"; (b) "A dog is laying with a remote controller"; (c) "A cube made of brick"; (d) "A red book and a yellow vase", from the original model and RL model respectively. Images in the same column are generated with the same random seed.

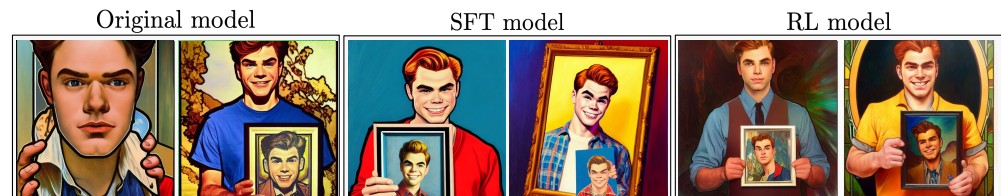

Figure 12: Text prompt: "oil portrait of archie andrews holding a picture of among us, intricate, elegant, highly detailed, lighting, painting, artstation, smooth, illustration, art by greg rutowski and alphonse mucha".

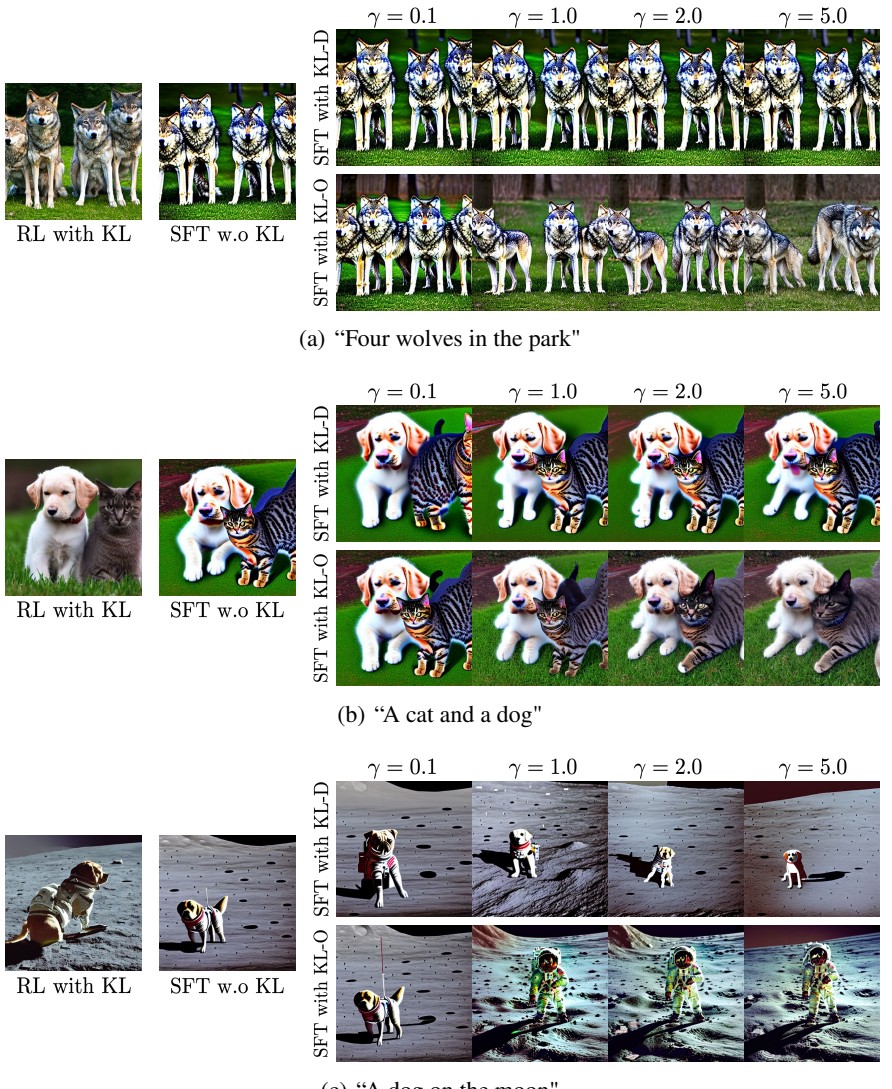

Figure 13: Samples from supervised fine-tuned models with different KL regularization and KL coefficients.

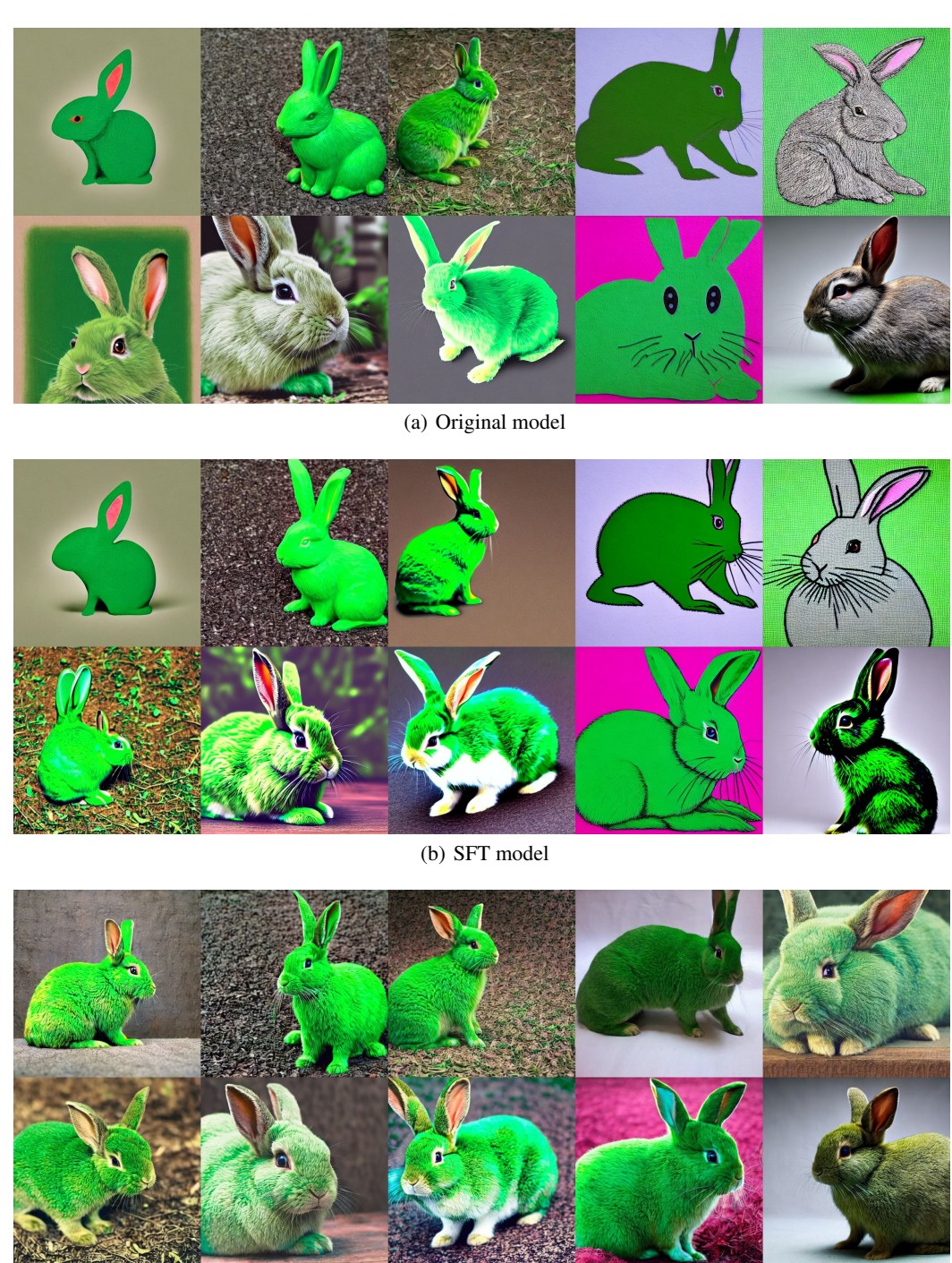

(a) Original model

(b) SFT model

(c) RL model

Figure 14: Randomly generated samples from (a) the original Stable Diffusion model, (b) supervised fine-tuned (SFT) model and (c) RL fine-tuned model.

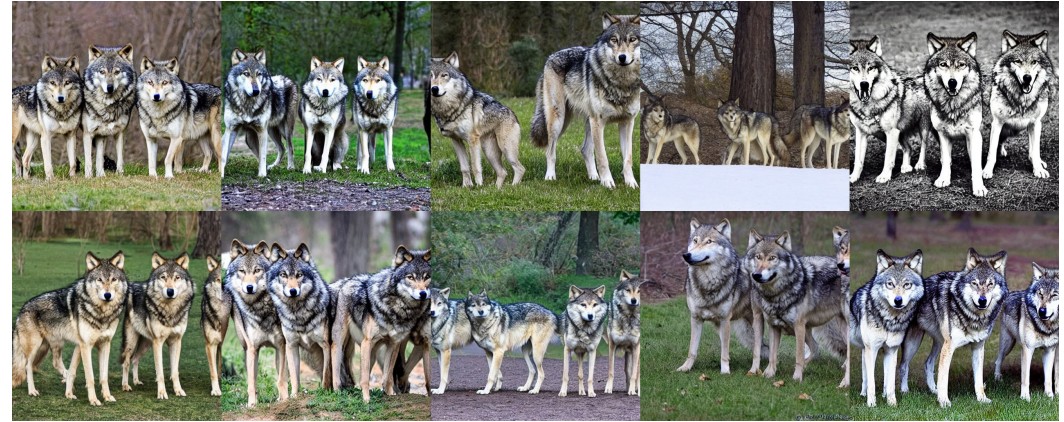

(a) Original model

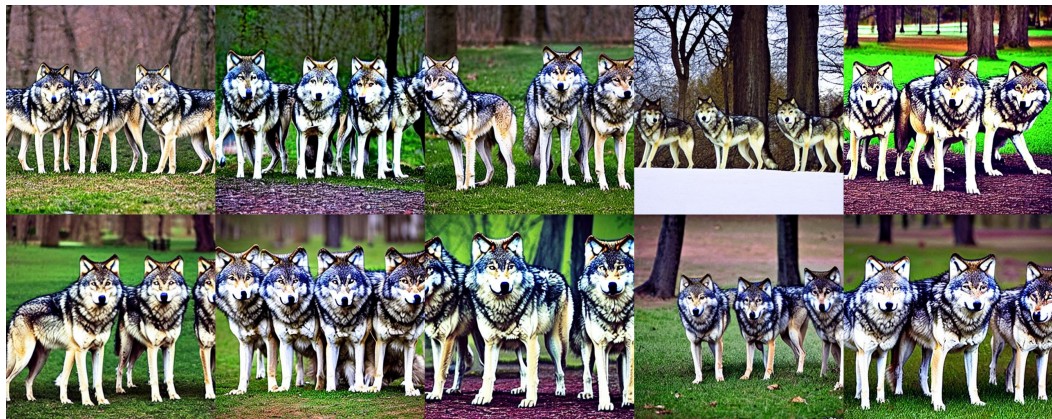

(b) SFT model

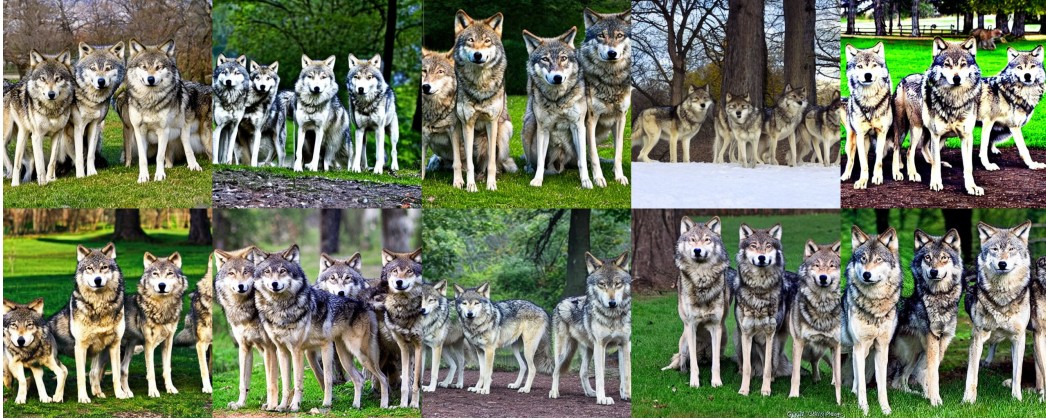

(c) RL model

Figure 15: Randomly generated samples from (a) the original Stable Diffusion model, (b) supervised fine-tuned (SFT) model and (c) RL fine-tuned model.

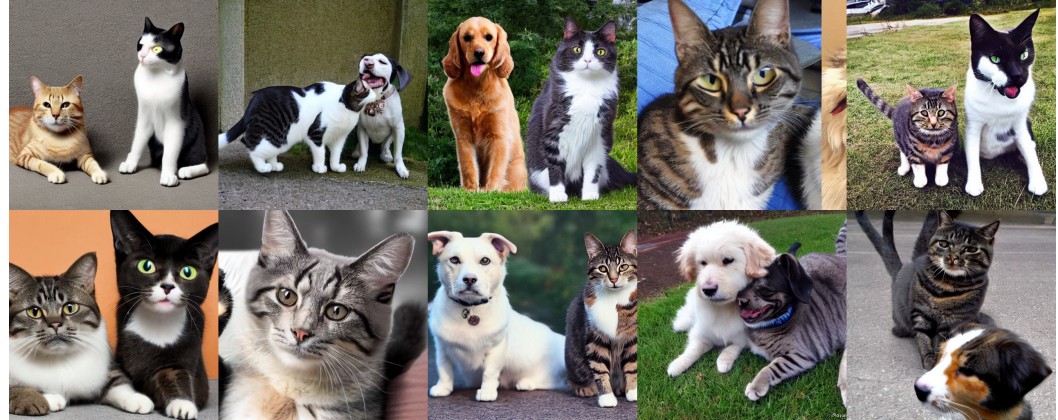

(a) Original model

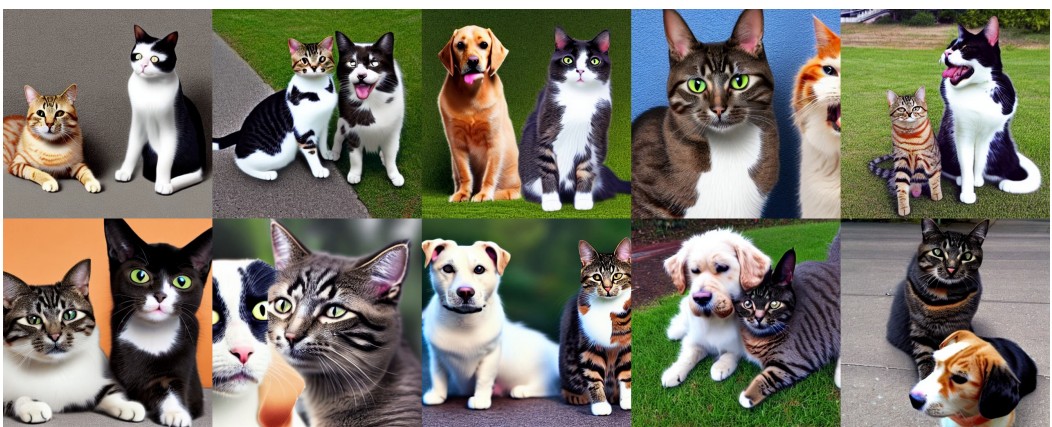

(b) SFT model

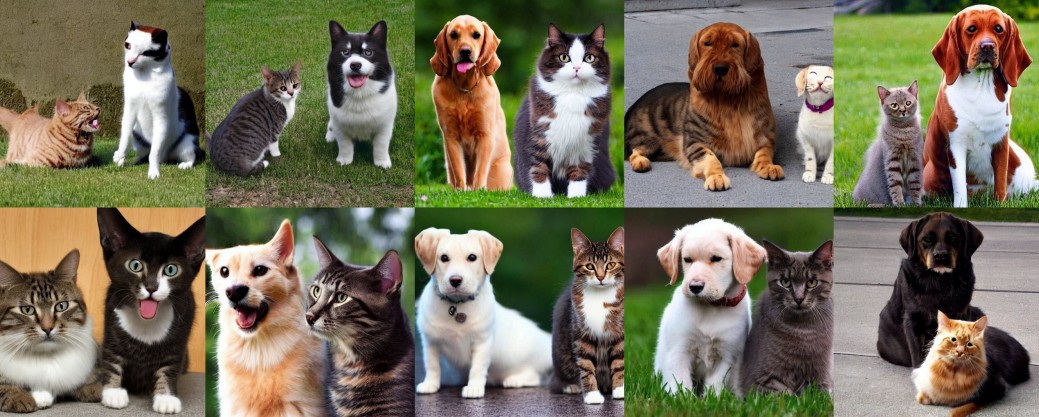

(c) RL model

Figure 16: Randomly generated samples from (a) the original Stable Diffusion model, (b) supervised fine-tuned (SFT) model and (c) RL fine-tuned model.

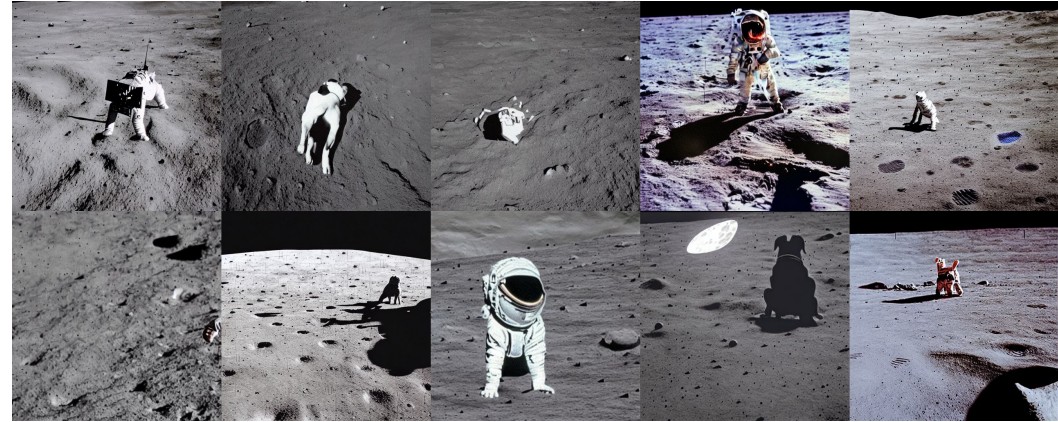

(a) Original model

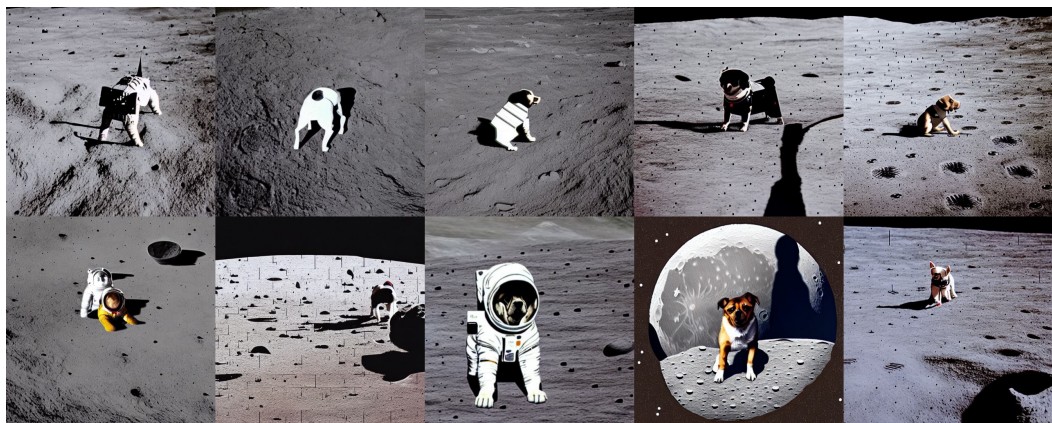

(b) SFT model

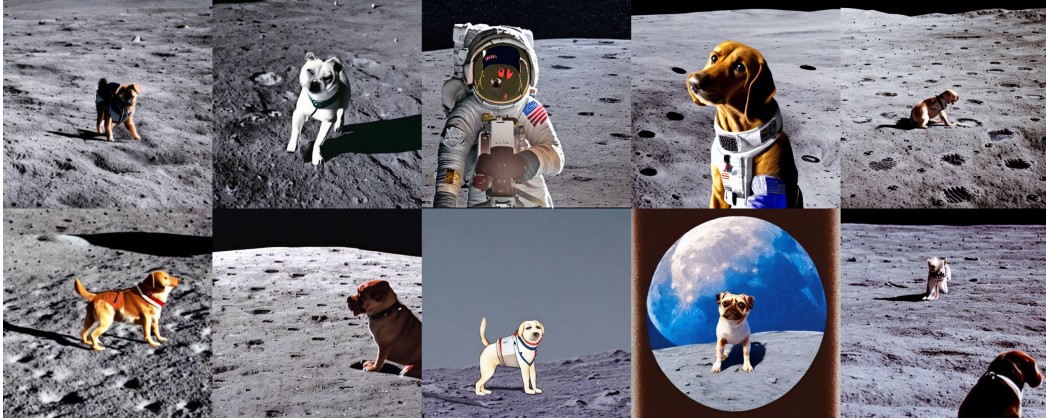

(c) RL model

Figure 17: Randomly generated samples from (a) the original Stable Diffusion model, (b) supervised fine-tuned (SFT) model and (c) RL fine-tuned model.

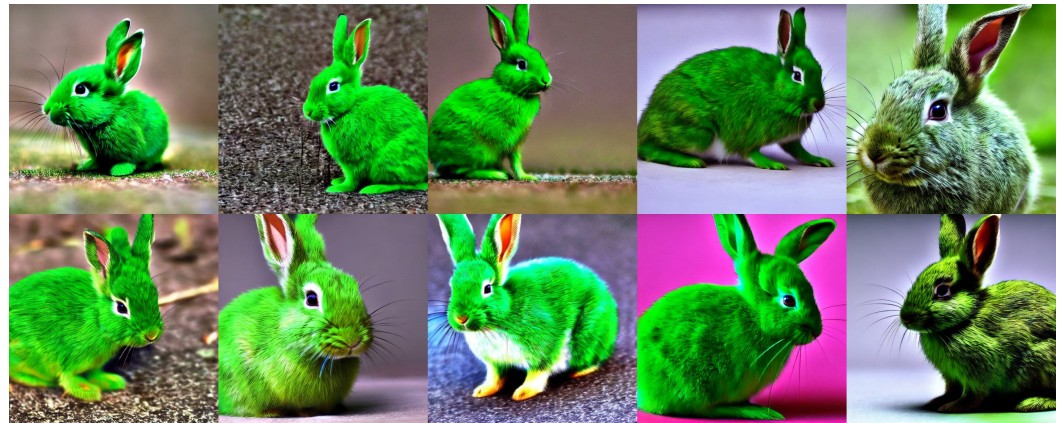

(a) RL model without KL regularization

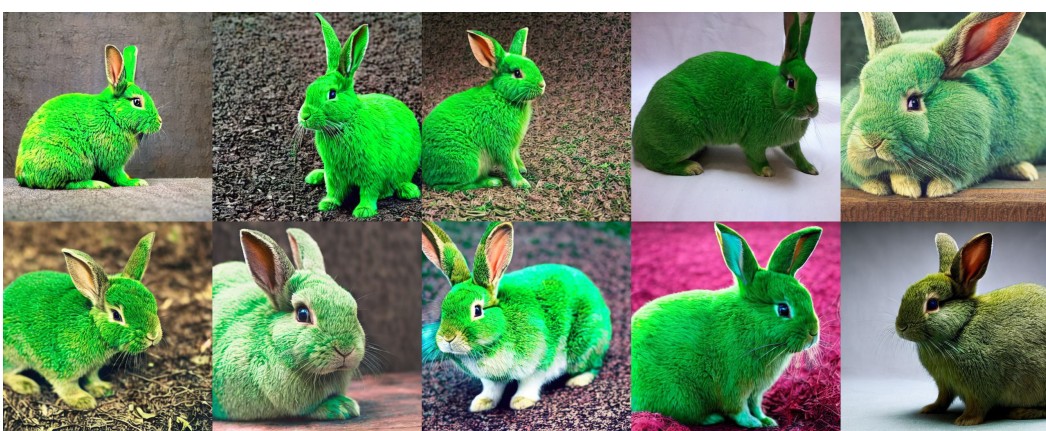

(b) RL model with KL regularization

Figure 18: Randomly generated samples from RL fine-tuned models (a) without and (b) with KL regularization.

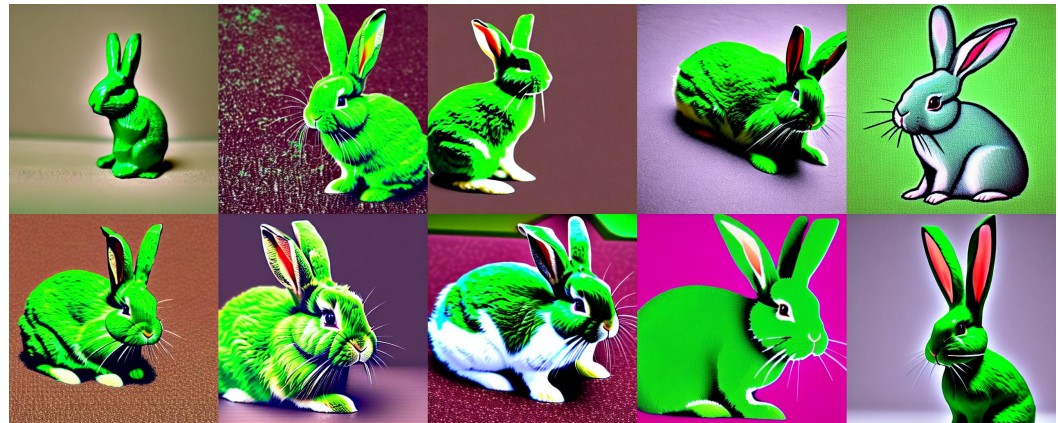

(a) SFT model without KL regularization

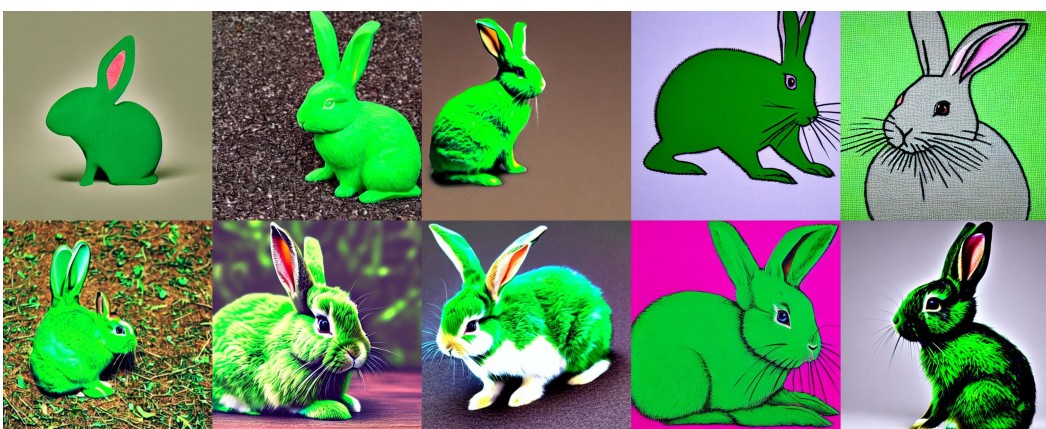

(b) SFT model with KL regularization

Figure 19: Randomly generated samples from supervised fine-tuned (SFT) models (a) without and (b) with KL regularization.

