# OpenReview forum: "DPOK: Reinforcement Learning for Fine-tuning Text-to-Image Diffusion Models"
_NeurIPS.cc/2023/Conference — NeurIPS 2023 poster_

### Official Review · Reviewer_gCGU · 2023-06-15

**Soundness:** 3 good
**Presentation:** 3 good
**Contribution:** 3 good
**Rating:** 5
**Confidence:** 4

**Summary:**

in this paper, the authors propose to improve the pretrained text-to-image diffusion model with human feedback in an online-reinforcement learning manner. The problem formulation, the differences between the online update and the supervised finetuning are clearly stated. Experiments show that the proposed method effectively improves the performance of the text-to-image generation ability.

**Strengths:**

1. The paper is well-written and easy to understand.

2. The comparison and the potential advantages compared to related works are clearly discussed.

3. Experiments show that the proposed method improves the performance of the pretrained text-to-image model, also outperforms the supervised human-feedback finetuning baseline.

**Weaknesses:**

1. Some claims and the results are connected weakly. The claims in Sec.4.3 should be better aligned with the experimental results. However, its current version makes me hard to find the direct mapping.

2. The title of this paper seems not to reflect the main contribution. In my opinion, the main contribution of this work lies in the online manner, since using reinforcement learning to fine-tune the text-to-image model has been explored before. It needs a better title.

3. There are some experimental problems. In my experience, the over-saturated problem in the SFT model can be alleviated by using small classifier-free guidance. Therefore, the necessity of using the online optimization is questionable. Also, why the authors do not directly compare with [17]?

**Questions:**

Please refer to the weakness part.

**Limitations:**

The authors discussed some limitations in the last part of the paper.

---

> ### Author Rebuttal · Authors · 2023-08-09
>
> We sincerely appreciate your valuable comments. We found them extremely helpful in improving our draft. We address each comment in detail, one by one below.
>
> **Comment 1. The connection between Section 4.3 and experiments:**
>
>
> It is hard to make a one-to-one mapping between each claim in Section 4.3 and the experimental results since Section 4.3 provides the overall combination of these differences as a comprehensive explanation of the benefits of using RL over SFT. Since RL and SFT are two very different methods with several differences, it could be hard to just select one difference and conduct a clean ablation study, due to the existence of other differences. Also, we do not claim that the superiority of RL over SFT is due to a single factor (or difference between them).
>
> **Comment 2. The necessity for online optimization:**
>
> First, **we propose online fine-tuning to better optimize ground truth rewards, and not just for solving the image quality degradation in fine-tuning**.
>
> Second, the over-saturated problem is just one aspect of downgraded image quality. There are other aspects that cannot be solved by adjusting the guidance weight (also note that only reducing the guidance weight could suffer from lower text-image alignment). In fact, fine-tuning with a limited distribution of images itself could suffer from worse image quality, so tricks like reconstruction loss have been proposed to solve such issues. Our KL regularization is performing a similar role.
>
> Finally, adjusting the classifier guidance scale did not make much difference for the SFT model in our experiments.
>
> **Comment 3. Comparison with [17]**
>
> **We did a comparison with [17] in our paper**, which corresponds to the SFT method without KL regularization. We added some theoretical justification and also tried some extra KL regularization methods on top of it.

---

> > ### Author Response · Authors · 2023-08-18
> >
> > Thank you again for the review and we hope that our individual response (https://openreview.net/forum?id=8OTPepXzeh&noteId=7cJZSAKzdA) and overall rebuttal (https://openreview.net/forum?id=8OTPepXzeh&noteId=UFj7ZsHm99) have addressed your main concerns. We would greatly appreciate your feedback and please feel free to let us know if you have any other questions in the discussion period!

---

> > > ### Comment · Reviewer_gCGU · 2023-08-19
> > >
> > > Thank the author for the rebuttal content. I have read all the contents and the reviews from other reviewers. There are 3 suggestions,
> > > + One is about the claim and the experiments, although it is hard to conduct a clean ablation. It will be more useful to show some generated examples to support the claim.
> > > + It is clear to cite the paper [17] in the experimental part.
> > > + The online manner is the main contribution, and I think the title may include this keyword.
> > >
> > > After careful consideration, I keep borderline accept to this paper.

---

> > > > ### Author Response · Authors · 2023-08-19
> > > >
> > > > Thank you for the reply!
> > > > >One is about the claim and the experiments, although it is hard to conduct a clean ablation. It will be more useful to show some generated examples to support the claim.
> > > >
> > > > As discussed with reviewer YMPp (https://openreview.net/forum?id=8OTPepXzeh&noteId=gDVthccGwl), **we already added some new results comparing offline and online RL methods, which shows that online RL is indeed better** as one ablation showing the differences between online and offline samples.  We will also add such ablation results in our draft.
> > > >
> > > > >It is clear to cite the paper [17] in the experimental part.
> > > > >The online manner is the main contribution, and I think the title may include this keyword.
> > > >
> > > > Thank you for the suggestions and we will modify them correspondingly in our draft.
> > > >
> > > > Please feel free to let us know if you still have any questions!

---

### Official Review · Reviewer_gRSW · 2023-06-28

**Soundness:** 3 good
**Presentation:** 3 good
**Contribution:** 3 good
**Rating:** 6
**Confidence:** 4

**Summary:**

The paper introduces the idea of finetuning text-to-image diffusion models using reinforcement learning. The core idea is relatively simple: generate samples from a trained diffusion model, use the samples and a reward function to update the model's parameters, and iterate. The paper further introduces a KL divergence loss to prevent diverging too far from the original model's weights. evaluations are done on different prompts with respect to different capabilities/ performance metrics, such as colour or composition.

**Strengths:**

- an interesting and novel idea
- simple method and easy algorithm
- qualitatively pleasing results

**Weaknesses:**

- considering only 4 prompts for experimental evaluation is not sufficient
- considering only a single prompt for bias evaluation is not sufficient
- authors mention that "longer training time, hyper-parameter tuning and engineering efforts are required as the number of training text prompts increases". This indicates that their method does not scale to a larger set of training prompts, which questions its relevance
- investigation of generalisation to prompts that are out of distribution should be conducted


**Questions:**

see weaknesses

**Limitations:**

limitations are briefly discussed, especially limitations mentioned in weaknesses could be discussed in more detail

---

> ### Author Rebuttal · Authors · 2023-08-09
>
> We sincerely appreciate your valuable comments. We found them extremely helpful in improving our draft. We address each comment in detail, one by one below.
>
> **Comment 1. Lacking experiments, training with more prompts, not scale to a larger set of training prompts**
>
> Thanks for your feedback. Please check our new results in the overall author rebuttal and global response pdf. In these documents, we have included experiments that contain training with a large variety of prompts, showcasing that **our method can also improve the average reward on much larger training sets with 104 MS-CoCo prompts and 183 Drawbench prompts (see Table 3 and Fig 1 in the response pdf).**
>
>
> **Comment 2. Generalization of out-of-distribution prompts:**
>
> **We conducted tests on unseen prompts, which consist of unseen objects, in Fig. 2 (b).** Notably, the fine-tuned models produced better images than the original model for these unseen text prompts. However, for better generalization, we need to train on a larger and more diverse set of text prompts.
>
> **We hope our response resolves all the concerns in your review, and please feel free to let us know if you have any other questions in the discussion period.**

---

> > ### Comment · Reviewer_gRSW · 2023-08-16
> >
> > Thank you for the remarks and the additional experiments run. I have updated my score accordingly.

---

### Official Review · Reviewer_VuBv · 2023-06-28

**Soundness:** 3 good
**Presentation:** 3 good
**Contribution:** 3 good
**Rating:** 5
**Confidence:** 3

**Summary:**

This paper proposes to use RL to finetune t2i diffusion models and use KL regularization into supervised fine-tuning of diffusion models.  The paper shows that RL fine-tuning can avoid the overfitting that arises in SFT, and is generally superior to SFT with respect to both image-text alignment and image quality.

**Strengths:**

The motivation for using RL techniques to fine-tune T2I model is reasonable and compelling.

The paper is well-written and presented. It is very easy to follow.

The idea is simple and easy to implement.

The proposed KL regularization technique is effective. The evidence provided in the paper supports the efficacy of this technique

**Weaknesses:**

The novelty of this paper is kind of limited. Similar ideas have already been proposed in [1].

I feel the experiment parts are a bit lacking. Only one policy gradient approach is tried in the paper and only 20K images are used for fine-tuning.

In the contribution part, the author claimed "online fine-tuning is critical to improving text-to-image alignment while maintaining high image fidelity", but fidelity scores or IS scores are not reported in the paper. Also no human evaluation are conducted in the main paper.

The cost for online RL fine-tuning is not reported. I believe it will be more cost-expensive compared with supervised fine-tuning, especially for scenarios such as fine-tuning on multiple prompts.


[1] Training diffusion models with reinforcement learning

**Questions:**

How do you define the aesthetic scores and the ImageReward scores? Do you use the same metric as in the Imagereward paper? I am also curious about the CLIP/BLIP score with the proposed method which are often used for the evaluation.

I think LoRA is proposed to supervised fine-tuning. Do you apply LoRA to both Supervised and RL fine-tuning? Also the UNet in the Stable Diffusion contains different layers including conv layers and cross-attention layers etc. Which part of the UNet module do you apply LoRA to?

**Limitations:**

Yes.

---

> ### Author Rebuttal · Authors · 2023-08-09
>
> We sincerely appreciate your valuable comments. We found them extremely helpful in improving our draft. We address each comment in detail, one by one below.
>
> **Comment 1. Novelty is limited, similar ideas have already been proposed in [1]**
>
> We respectfully disagree with the reviewer’s assessment. In fact, it is unfair to list the similarity to DDPO [1] as a weakness of our work, because 1) our work and DDPO [1] were developed independently, and 2) DDPO is an unpublished work, which appeared online only after the NeurIPS-2023 submission deadline. Moreover, we study the effect of KL-regularization (whose importance in fine-tuning foundation models has been well-established in the literature) in both online and supervised learning approaches, supported by theoretical justifications and experimental results, which has not been explored in the DDPO paper [1].
>
>
> [1] Training diffusion models with reinforcement learning
>
> **Comment 2. Lacking experiments and fidelity score**
>
> Thanks for your feedback. Please check the overall author rebuttal and global response pdf. In these documents, we have included experiments that contain training with a large variety of prompts, showcasing that **our method can also improve the average reward on 104 MS-CoCo prompts and 183 Drawbench prompts (see Table 3 and Fig 1 in the response pdf).** Furthermore, we have included **human evaluation results on a single prompt (see Tables 1&2 in the response pdf)**, which show that our RL model outperforms the SFT model in terms of both alignment and image quality. Moreover, our RL model outperforms the original model in terms of alignment, while maintaining comparable image quality.
>
> As mentioned in the paper, we use the aesthetic predictor from LAION [2] to measure the image quality (fidelity), since we select some synthetic and challenging prompts for which we do not have ground truth images to compute FID.
>
> [2] Schuhmann et al. Laion-5b: An open large-scale dataset for training next generation image-text models. arXiv preprint arXiv:2210.08402, 2022.
>
> **Comment 3. The cost for training:**
>
> **We reported the cost for RL training in Appendix B**. It is true that RL training requires a relatively smaller learning rate, making it computationally more expensive. However, the increase in computational requirements is not significantly large as we also need a smaller learning rate for supervised fine-tuning to achieve higher image quality.
>
> **Comment 4. ImageReward**
>
> Yes, as mentioned in our paper, we use the same way to evaluate ImageReward and aesthetic scores as in the ImageReward paper. Also, it has been demonstrated that ImageReward is more correlated with real human decisions than CLIP/BLIP. As a result, we believe that the ImageReward score offers a more meaningful and relevant signal for evaluation purposes.
>
> **Comment 5. LoRA fine-tuning**
>
> We use LoRA fine-tuning for both RL and SFT. Also, we applied LoRA to the cross-attention layers.

---

> > ### Author Response · Authors · 2023-08-18
> >
> > Thank you again for the review and we hope that our individual response (https://openreview.net/forum?id=8OTPepXzeh&noteId=pZZwGatqVk) and overall rebuttal (https://openreview.net/forum?id=8OTPepXzeh&noteId=UFj7ZsHm99) have addressed your main concerns. We would greatly appreciate your feedback and please feel free to let us know if you have any other questions in the discussion period!

---

> > > ### Comment · Reviewer_VuBv · 2023-08-18
> > >
> > > Thanks for the authors' efforts spending on the rebuttal! Similar to reviewer fxhn and YMPp, I still have concerns about the reward used in this paper for RL training. Moreover, even though the paper conducts LoRA fine-tuning for both the RL setting and SFT, there is no insight given on these methods.. In my view, using LoRA for the RL setting is completely different and may bring additional uncertainty. Overall, I read all responses and I would like to keep my score.

---

> > > > ### Author Response · Authors · 2023-08-18
> > > >
> > > > Thanks for your reply!
> > > >
> > > > 1) Could you elaborate on your concern about the reward used in the paper? About **Reviewer fxhn**'s concern on extra evaluations besides ImageReward and aesthetic score, in our rebuttal **we added new human evaluation**; We also show that **training with reward in [17] also works well for our RL method** since **Reviewer YMPp** asked about training with other reward choices, and we also include a discussion on our method with KL regularization will help in our RLHF setting, etc (https://openreview.net/forum?id=8OTPepXzeh&noteId=8z8kat5mBG). **We believe that we have addressed the above concerns and it would be extremely helpful if you could point out what is still unclear to you!**
> > > >
> > > > 2) As in the rebuttal to Reviewer fxhn (Comment 3. LoRA fine-tuning vs. full fine-tuning), we mentioned that "**In fact, we first tried fine-tuning with all parameters which also works well, but we finally switched to LoRA fine-tuning for more efficient training**". **In our experiments, fine-tuning with or without LoRA does not make a noticeable difference except for the GPU occupation**.
> > > >
> > > > [17] Aligning Text-to-Image Models using Human Feedback. Kimin Lee, Hao Liu, Moonkyung Ryu, Olivia Watkins, Yuqing Du, Craig Boutilier, Pieter Abbeel, Mohammad Ghavamzadeh, Shixiang Shane Gu

---

### Official Review · Reviewer_YMPp · 2023-07-04

**Soundness:** 3 good
**Presentation:** 3 good
**Contribution:** 2 fair
**Rating:** 5
**Confidence:** 3

**Summary:**

This paper studies RLHF fine-tuning of diffusion models to learn from and align with human feedback. Specifically, they introduce (1) an online RL strategy and (2) a KL regularization (inspired by a similar regularization for the RLHF via PPO of LLMs). Empirically, they show better alignment with the optimized reward and increased aesthetic when compared to the more trivial offline imitation based RL (named SFT in the paper).

**Strengths:**

* The paper is clear, the theoretical section is sound, and the experiments are convincing.
* This is arguably the first work applying online RL to fine-tune diffusion model, which is a key contribution.
* I appreciated the MDP formulation of the diffusion process.
* The inclusion of the KL divergence is interesting, though straightforward given the recent success of RLHF in NLP.

**Weaknesses:**

* Section 4.3 proposes 3 theoretical/intuitive reasons why online RL would perform better than SFT. I believe those reasons could be further validated in the experiments.
    - The first argument is that online learning favors exploration away from the pre-trained distribution. This could be ablated by saving the generated images along the online RL, and then applying SFT on these images.
    - The second argument is the difference in KL regularization, which actually theoretically comes to the difference between KL and reverse KL. Applying online RL with reverse KL (though more costly) could help ablate the importance of this component. Actually, I am sceptikal by the KL-D for SFT, as (1) the more standard KL-0 in App.E actually performs better and (2) offline RL usually requires baseline rewards for normalization, thus adjusting the original reward with a shift factor would not change anything.
    - The third argument is the robustness of the reward model. Therefore, ablations could better analyze the quality difference of the reward model on the pre-train distribution and then on the updated distribution. More generally, the fact that your online RL requires a more robust reward model is actually a drawback/limitation, creating new challenges in robust reward design.
* More generally more experiments are required to ensure that online RL for diffusion models consistently helps.
    - The experiments are made only for a very small number of prompts: 4! This limits applications to real-world applications.
    - The experiments are made with a single reward model. More experiments with diverse (less robust) reward models would help.
    - Therefore the empirical contribution may be seen as marginal.
    - The paper lacks human evaluation, even though this was done for the reward model. I believe human evaluation after online RL vs SFT would have been more interesting.

**Questions:**

* Could you elaborate on the difference between KL and reverse KL.
* Do you think it may be possible to add "online" in the title "Reinforcement Learning for Online Fine-tuning of Text-to-Image Diffusion Models", to specify that your contribution is actually about online RL.
* I believe your SFT baselines could be improved with some reward normalization, value function and with KL-D. Such improved SFT baseline would strengthen your experimental sections.
* Have you explored refined online RL optimization strategies such as PPO?
* Finally, I believe more prompts (for example those used in evaluation) and more reward models (for example those used for aesthetic evaluation) are needed in training.

**Limitations:**

The limitations could be extended to the societal risks of generative AI, and also mention the fact that online RL requires more robust reward models.

---

> ### Author Rebuttal · Authors · 2023-08-09
>
> We sincerely appreciate your valuable comments. We found them extremely helpful in improving our draft. We address each comment in detail, one by one below.
>
> **Comment 1. Ablation study to verify the effect of exploring online samples**
>
> Response 1: This is an interesting question! However, we notice that **online/offline is not the only difference between RL & SFT training**: noticing that weighted ELBO (the RHS of Eq. (9)) is equivalent to a weighted score matching loss in [35], SFT is learning the score function with high rewards from a fixed denoising process, where RL model explores the trajectories with high rewards. As a result, even if we use SFT to train on images generated by the RL model, there are still other differences between such SFT and the RL training.
>
> In fact, **RL & SFT are two very different methods with many different aspects, and the first difference discussed in Section 4.3 is only one aspect**, so it could be hard to just select one aspect and conduct a clean ablation study due to other differences. One possible approach is to conduct offline RL by collecting trajectories generated from the original model and comparing it to the online RL, but it would not be a comparison between SFT and RL. We also expect that such an offline method would be outperformed by the online method. This is because the performance of the offline method is upper-bounded by the data coverage, and thus, online exploration would improve the model performance when the offline data coverage is not good. Finally, we want to mention that even if we intuitively treat SFT as an offline method, it is not an offline RL method, which is also discussed in Response 2.
>
> **Comment 2. Question about KL-D for SFT:**
>
> Response 2: Thank you for the question, but we think that your claim “the shift in the reward would not change anything in SFT” is not accurate. We would like to highlight that **there is no policy gradient in SFT**, and **subtracting a baseline without changing the expectation of the gradient estimation only applies to policy gradient methods**. Actually, the solution of the weighted ELBO (the RHS of Eq. (9), equivalent to the score matching loss in [35], which is an L2-loss between the actual score and the score function model) could change when $\gamma$ changes: under the assumption that reward is non-negative, the lowest possible reward is 0. If there are images with 0 reward in the SFT dataset, we will give 0 weight to the score matching loss of such images. If we add a positive shift, the reward will be non-zero and it will give a positive weight on learning the score function of such images.
>
> Empirically, as we can observe in Fig 7(a), increasing $\gamma$ will result in a lower reward for SFT training. Moreover, since SFT is not a policy gradient method, it will not benefit from baseline/value function tricks. For more results in KL-D, we present our full ablations result in Section E in the appendix.
>
> **Comment 3. The robustness of the reward model:**
>
> Response 3: To clarify, the third difference mentioned in Section 4.3 is under the assumption that we have a good reward model. Our point is that if we have a reward model which generalizes well or is even perfect, **SFT will not fully utilize that model by only evaluating the reward on a fixed dataset, whereas RL has the capability to leverage it more comprehensively**.
>
> Also, we consider the case when the reward model could be imperfect, which is **why we add KL-regularization to avoid over-optimization and out-of-distribution reward evaluation**.
>
> **Comment 4. More experiments with multiple prompts and human evaluations:**
>
> Response 4: Thanks for your feedback. Please check the overall author rebuttal and global response pdf. In these documents, we have included experiments that contain training with a large variety of prompts, showcasing that **our method can also improve the average reward on 104 MS-CoCo prompts and 183 Drawbench prompts (see Table 3 and Fig 1 in the response pdf).** Furthermore, we have included **human evaluation results on a single prompt (see Table 1&2 in the response pdf)**, which demonstrate that our RL model outperforms the SFT model in terms of both alignment and image quality. Moreover, our RL model outperforms the original model in terms of alignment while maintaining comparable image quality.
>
> **Comment 5. Elaborate on the difference between KL & reverse KL:**
>
> Response 5: The key difference is which distribution is used to evaluate the difference in the log probability. In KL for SFT, we evaluate using the offline distribution, which is the training data; in KL for RL, we evaluate using online generated trajectories.
>
> **Comment 6. Minor issue about the title**
>
> Response 6: Thanks for pointing it out! We will address this in the final draft.
>
> **Comment 7. Refined online RL optimization like PPO**
>
> Response 7:  As mentioned in Appendix B, we tried a small learning rate and clipping the gradient norm to stabilize the training, which is similar to PPO. We also tried importance sampling and clipping the ratio, where we did not find much improvement in the learning process. We will clarify this in the final draft.

---

> > ### Comment · Reviewer_YMPp · 2023-08-15
> >
> > Thanks for the clarifications and rebuttal. You state that "One possible approach is to conduct offline RL by collecting trajectories generated from the original model and comparing it to the online RL"; that would indeed be necessary to validate that online RL performs better than offline RL. Moreover, I still think that more ablation studies and additional experiments on diverse rewards is necessary. Overall, I read all responses and decided to keep my score.

---

> > > ### Author Response · Authors · 2023-08-16
> > >
> > > Thank you for your additional comments!
> > >
> > > **Q1. Offline RL vs Online RL**
> > >
> > > Following your suggestion, we fine-tune the model using offline RL: for the prompt “A green colored rabbit”, we employ an advantage-weighted regression as an offline policy learning algorithm [2] with trajectories generated by the original model. Specifically, we use an exponential weighted advantage objective that learns a policy maximizing the Q-values subject to a distribution constraint (Eq.(7) in [2]) and try different $\beta$ parameters (i.e., $\beta \in 1,2,3,10$). In our experiment, **offline RL only improves the average reward from ~0.1 to ~0.3 (after 5000 gradient updates), while our online RL approach can quickly improve the reward to ~1.5 before 5000 gradient updates.** Because offline RL is usually unstable and its performance is upper-bounded by data coverage, we find that offline RL is worse than online RL. We hope that this additional result can clarify your question.
> > >
> > > [2] Offline Reinforcement Learning with Implicit Q-Learning. Ilya Kostrikov, Ashvin Nair, Sergey Levine
> > >
> > > **Q2. Ablation study and rewards**
> > >
> > > Would you be able to provide more details regarding the specific ablation studies and reward functions that we should consider? We are committed to addressing this concern to the best of our ability.
> > >
> > > If you have any other questions or suggestions, please do not hesitate to let us know.

---

> > > > ### Comment · Reviewer_YMPp · 2023-08-16
> > > >
> > > > Thank you for this additional and informative baseline. Many rewards could be considered (not only aesthetic ones, eg, nsfw-filtering ones or personalized ones), and all may not be robust. And as stated in the initial review: "the fact that your online RL requires a more robust reward model is actually a drawback/limitation, creating new challenges in robust reward design". Thus experiments with rewards weaker than ImageReward may be interesting to show when online learning actually succeeds, and in what situations the proposed method might fail.

---

> > > > > ### Author Response · Authors · 2023-08-19
> > > > > **Clarification on reward models by authors**
> > > > >
> > > > > Thank you for your reply!
> > > > >
> > > > > In our initial experiments, we actually used a weak reward model from [17], which is trained on small-scale human datasets compared to ImageReward. Our online RL fine-tuning worked well on tested text prompts, surpassing the performance of supervised fine-tuning. **This result shows the robustness of our method on weak reward function**. However, the reward model in [17] is only trained with a limited category of prompts, so we adopt ImageReward for a wider range of prompts.
> > > > >
> > > > > We agree that the robustness of the reward model is important in the RLHF framework. However, **learning a reward function from human feedback has shown to be effective and robust in many prior works**; Besides, there is some additional component like **KL regularization** to mitigate the issues from reward models, which is also adopted in our work.
> > > > >
> > > > > Thank you for your valuable comments on reward models! We leave the exploration of other reward choices as future work since our current version mainly focus on the RLHF setting, and we will add related discussions in the final version. Please let us know if you still have other questions!
> > > > >
> > > > > [17] Aligning Text-to-Image Models using Human Feedback. Kimin Lee, Hao Liu, Moonkyung Ryu, Olivia Watkins, Yuqing Du, Craig Boutilier, Pieter Abbeel, Mohammad Ghavamzadeh, Shixiang Shane Gu

---

### Official Review · Reviewer_fxhn · 2023-07-06

**Soundness:** 3 good
**Presentation:** 3 good
**Contribution:** 3 good
**Rating:** 5
**Confidence:** 4

**Summary:**

The paper proposes to adapt the popular RLHF framework used for fine-tuning LLMs to fine-tuning of diffusion generative models.
Given a reward model, the goal is to finetune the parameters of the diffusion model such that the resulting images from the
sampling process achieve high reward. Importantly, the reward is a function of the final output of the entire diffusion process (online finetuning) and can thus not be optimized independently across timesteps like the denoising loss used to train the LDM. The authors therefore adapt a previous result to compute the gradient without the need of backpropagating through the entire diffusion graph. Additionally, they regularize the optimisation via an upper bound on the KL divergence between the fine-tuned and the original model.

**Strengths:**

 - I find the idea to adapt RLHF from language tasks to diffusion generative models to be very relevant as diffusion models clearly often misinterpret details like counts and colors about the prompt.
 - It is good that the authors added KL regularization to supervised finetuning to establish a better baseline for their RLHF framework.
 - Section 4.3 clearly distinguishes RLHF from supervised finetuning and gives several good reasons why RLHF is superior to supervised finetuning from a more theoretical perspective.
 - I find the quality of the results in the paper to be mostly convincing and I get the impression that RLHF significantly outperforms both baselines.

**Weaknesses:**

 - The paper focuses mostly on finetuning on a single or few prompts at a time. Being able to train on a large variety of prompts to train a model that is overall better than SD instead of only focusing on a few specific aspects would greatly increase the usefulness of the paper.
 - The evaluation is somewhat limited and it is not clear how cherry-picked the prompts for the results are (for the given prompts the appendix contains non-cherry-picked results). It would be much more convincing if the authors could show results on challenging prompts that are randomly generated. Also, the quantitative results in Figure 3 would be more impressive if they were averaged over a larger selection of prompts.
 - Since the method is trained to improve ImageReward, I would prefer a different evaluation metric in Figure 3 to demonstrate generalization. If there is no automatic pipeline available, the authors could conduct a user study or simply generate prompts that are easy to evaluate by a human. For example, prompts of the form "{N} {CLASS} {BACKGROUND}" , where N is the number of objects and CLASS the object to generate. Then they could simply generate a certain number of images and manually count the ones where the model correctly generates the required number of objects.
 - In general, the evaluation is in my opinion the weakest part of the paper and I would gladly increase my score if the authors could present more convincing quantitative evaluations or a broader selection of qualitative results (or both).
 - While RLHF is significantly better than the baselines, to me some of the images, especially the "green rabbit" and "green cat" still look somewhat oversaturated.

**Questions:**

 - Is there a reason why eq (8) should not include a min, similar to (4)?
 - Do you believe that LoRA helps achieve better results or would standard finetuning on the original dimensional weights yield better results if compute would not be a concern?
 - From my experience, diffusion models have issues with multiple colors in a single prompt. For example a "green rabbit and a yellow bird  in front of a blue background". You show that RLHF can improve colors but can it also handle multiple colors. Also diffusion models often fail with text, for example "a tshirt with NeurIPS printed on it", does RLHF help with this or does the reward function not yield useful feedback for this kind of prompt?

**Limitations:**

The main limitation of the paper is the restriction to a single prompt which is openly discussed by the authors.

---

> ### Author Rebuttal · Authors · 2023-08-09
>
> We sincerely appreciate your valuable comments. We found them extremely helpful in improving our draft. We address each comment in detail below.
>
> **Comment 1. Fine-tuning on a single or few prompts at a time**
>
> Thanks for your suggestion! We added experiments that contain training with a large variety of prompts in global response pdf. The results show that **our method can improve the average reward on 104 MS-CoCo prompts and 183 Drawbench prompts**. For details, please check the overall author rebuttal and global response pdf (**Table 3 and Fig 1 in the response pdf**).
>
> **Comment 2. Human evaluation**
>
> Thanks for your suggestion! We conducted **human evaluation as a supplement of Fig 3**  in global response pdf. In our human evaluation, **the RL model consistently outperforms the SFT model in terms of both alignment and image quality and also outperforms the original model in terms of alignment with comparable image quality**. For details, please check the overall author rebuttal and global response pdf (**Table 1&2 in the response pdf**).
>
> **Comment 3. LoRA fine-tuning vs. full fine-tuning**
>
> In fact, we first tried fine-tuning with all parameters which also works well, but we finally switched to LoRA fine-tuning for more efficient training.
>
> **Comment 4. Multiple colors & text generation**
>
> This is an interesting question! We believe the answer to the question “whether RLHF helps with such tasks” depends on the capability of the reward function. In the new experiments with multiple prompts, we find that **RL fine-tuning leads to enhanced reward scores (from 0.76 to 1.57) on text prompts with multiple colors (e.g., yellow vase and red book) and produces better-aligned images compared to the original model (see Fig 3(d) in the response PDF).** Regarding text rendering, although the reward scores show improvement (from 0.45 to 1.11), there is no substantial difference in the generated images. However, we expect that with a more sophisticated reward function that is capable of capturing finer details, RLHF can improve text rendering.
>
> These results support our claim that RLHF remains effective even in highly challenging scenarios, encompassing multiple colors and text rendering.
>
> **Comment 5. Minor issues like the adding min in eq(8)**
>
> Thanks for pointing them out! We will fix them in the draft.

---

> > ### Author Response · Authors · 2023-08-18
> >
> > Thank you again for the review and we hope that our individual response (https://openreview.net/forum?id=8OTPepXzeh&noteId=usVNThWza4) and overall rebuttal (https://openreview.net/forum?id=8OTPepXzeh&noteId=UFj7ZsHm99) have addressed your main concerns. We would greatly appreciate your feedback and please feel free to let us know if you have any other questions in the discussion period!

---

### Author Rebuttal · Authors · 2023-08-09

Overall author rebuttal:

We thank all reviewers for their thoughtful comments. We greatly appreciate all the reviewers' acknowledgment that our method is **empirically effective with solid theories**. To address common concerns about scaling up the training prompts and human evaluation, we have add new experiments and evaluations:

**1. Training with a large variety of prompts:**

We conduct online RL training with 104 MS-CoCo prompts and 183 Drawbench prompts, respectively (the prompts are randomly sampled during training, and the full prompt dataset will be made public). We use the same configuration in Appendix B but with a longer training time till it generates 10K online samples (50K gradient steps). We report both ImageReward and the aesthetic score of the original and the RL fine-tuned models. For evaluation, we generate 30 images from each prompt and report the average scores of all images. **The evaluation result is reported in Table 3 with sample images in Fig 1 in the author rebuttal pdf, showing that RL training can also significantly improve the ImageReward score while maintaining a high aesthetic score with much larger sets of training prompts**.

**2. Human evaluation:**

As a supplementary evaluation to Fig. 3, we conduct extra human evaluation. We gather 40 images that are randomly generated from each prompt (“green rabbit”, “cat and dog”, “dog on the moon” and “four wolves”), resulting in a total of 160 images from each model (RL, SFT and the original model). Given two (anonymized) sets of four images from the same random seeds, one from RL fine-tuned model and one from the original model (or SFT model), we ask human raters to assess which one is better w.r.t. image-text alignment and fidelity (i.e., image quality). Each query is evaluated by 8 independent human raters and we report the average win/lose rate from RL&SFT comparison and RL&original-model comparison. **The results are presented in Table 1 & 2 in the author rebuttal pdf, showing that the RL model consistently outperforms the SFT model on both alignment and image quality and also outperforms the original model in the sense of alignment with comparable image quality**.

We will also include all the results in the final draft.

---

### Author Response · Authors · 2023-08-21

Dear AC and reviewers:

Thank you so much for your time and the engagement in the discussion! We summarize the **extra problems raised by reviewers and our overall response** as follows:

**1) Ablation study to verify the claim in Section 4.3 about online vs. offline samples**:

**We conduct experiments to compare online and offline RL methods** (which is not the SFT method used in the original submission, see details here: https://openreview.net/forum?id=8OTPepXzeh&noteId=gDVthccGwl), which shows that **the online RL approach improves the average reward much faster than the offline approach**. It is not surprising since the performance of the offline approach is upper-bounded by data coverage.

**2) Ablation study to use other reward models in RL training**:

**Weaker rewards from human feedback**: In our initial experiments, we also used a weak reward model from [17], which is trained on a smaller-scale human dataset compared to ImageReward. Our online RL fine-tuning also worked well on tested text prompts, surpassing the performance of supervised fine-tuning. **This result shows the robustness of our method on weaker reward functions trained with human comparison.** Besides, we propose online KL regularization designed for diffusion models, showing that it is also useful to mitigate the issue of evaluating out-of-distribution samples using the reward and improve the image quality.

[17] Aligning Text-to-Image Models using Human Feedback. Kimin Lee, Hao Liu, Moonkyung Ryu, Olivia Watkins, Yuqing Du, Craig Boutilier, Pieter Abbeel, Mohammad Ghavamzadeh, Shixiang Shane Gu

**For other reward functions**: Our work mainly focuses on the RLHF setting (where the reward function is trained with human comparison like ImageReward): The main reason is that this kind of reward is a good fit for policy gradient methods, since it is not a sparse reward and has shown great success in RLHF in the language domain. It is well-known that policy gradient would fail with a very sparse reward, so it is not our claim to train with arbitrary reward functions, but it could be interesting for future exploration.

We believe that we have resolved the extra issues raised by the reviewers during the discussion period, and we will include all the results and necessary discussions in the revised draft. We also believe that our proposed method of online fine-tuning of diffusion models with KL regularization provides meaningful insight and important contribution to the community.

Best,

Authors

---

### Decision · Program_Chairs · 2023-09-21

**Decision:**

Accept (poster)

**Comment:**

This paper proposes a reinforcement learning from human feedback (RLHF) method for diffusion models on text-to-image generation.
From the method innovation perspective, RLHF for diffusion models is a novel idea; it is an "obvious" but meaningful next step for text-to-image generation given the success of RLHF for large language models (LLMs). The reviewers acknowledged the significance of this work and praised the use of KL divergence. They unanimously lean toward acceptance of this work (most are borderline accept though). The AC carefully read the rebuttal and author-reviewer discussions, and feels that most concerns are addressed in the rebuttal. One reviewer mentioned another concurrent work that was released after the NeurIPS deadline, which, however, should not invalidate this paper's novelty and contributions at all. Overall, this is a meaningful paper with novel method and promising results, despite some limitations. The AC tend to accept this work. The authors are encouraged to include the additional results and human evaluation into the revision.